# ScienceBoard:
# Evaluating Multimodal Autonomous Agents in Realistic Scientific Workflows

**Qiushi Sun**[♡] **Zhoumianze Liu**[◇ℂ] **Chang Ma**[♡] **Zichen Ding**[◇] **Fangzhi Xu**[◇] **Zhangyue Yin**[ℂ]
**Haiteng Zhao**[☆] **Zhenyu Wu**[◇] **Kanzhi Cheng**[♣] **Zhaoyang Liu**[◇] **Qintong Li**[♡]
**Jianing Wang**[♠] **Xiangru Tang**[✆] **Tianbao Xie**[♡] **Xiachong Feng**[♡] **Xiang Li**[♠]
**Ben Kao**[♡] **Wenhai Wang**[◇] **Biqing Qi**[◇] **Lingpeng Kong**[♡] **Zhiyong Wu**[◇]
[♡]The University of Hong Kong [◇]Shanghai AI Laboratory [ℂ]Fudan University
[☆]Peking University [♣]Nanjing University [♠]East China Normal University [✆]Yale University
`{qiushisun,changma}@connect.hku.hk, {kao,lpk}@cs.hku.hk`
`{liuzhoumianze,wangwenhai,qibiqing,wuzhiyong}@pjlab.org.cn`

## Abstract

Large Language Models (LLMs) have extended their impact beyond Natural Language Processing, substantially fostering the development of interdisciplinary research. Recently, various LLM-based agents have been developed to assist scientific discovery progress across multiple aspects and domains. Among these, computer-using agents, capable of interacting with operating systems as humans do, are paving the way to automated scientific problem-solving and addressing routines in researchers' workflows. Recognizing the transformative potential of these agents, we introduce ScienceBoard, which encompasses two complementary contributions: (i) a realistic, multi-domain environment featuring dynamic and visually rich scientific workflows with integrated professional software, where agents can autonomously interact via different interfaces to accelerate complex research tasks and experiments; and (ii) a challenging benchmark of 169 high-quality, rigorously validated real-world tasks curated by humans, spanning scientific-discovery workflows in domains such as biochemistry, astronomy, and geoinformatics. Extensive evaluations of agents with state-of-the-art backbones (*e.g.*, GPT-5, Claude-Opus-4.6, UI-TARS) show that, despite some promising results, they still fall short of reliably assisting scientists in complex workflows, achieving only a 20% overall success rate. In-depth analysis further provides valuable insights for addressing current agent limitations and more effective design principles, paving the way to build more capable agents for scientific discovery. Our code, benchmark, and leaderboard are available at Scienceboard Homepage.

## 1 Introduction

In the pursuit of scientific advances, researchers combine ingenuity and expertise to perform novel research grounded in experimental explorations. In the modern era, scientific discovery is increasingly driven by specialized software and tools that empower scientists to engage deeply with the experimental world (Hacking, 1983). Tools like simulation engines (Hollingsworth & Dror, 2018), data analysis software (The MathWorks Inc., 2022), and visualization platforms (Goddard et al., 2018) are essential for formulating hypotheses, validating results, and advancing scientific understanding.

However, as scientific software grows more sophisticated and workflows become more demanding, the learning curve and operational burden on human researchers intensify (Sänger et al., 2024). These challenges motivate the vision of autonomous agents to play a central role in automating research pipelines and assisting human researchers as "AI co-scientists" (Luo et al., 2025; Schmidgall et al., 2025; Gottweis et al., 2025). For example, while a human scientist may take weeks to master a protein analysis tool (Meng et al., 2023) and spend hours making sufficient observations, an autonomous agent could perform the same tasks within minutes. By enabling fully autonomous workflows—from tool usage to making novel discoveries (Lu et al., 2024a)—such agents promise to accelerate science and empower researchers with unprecedented capabilities.

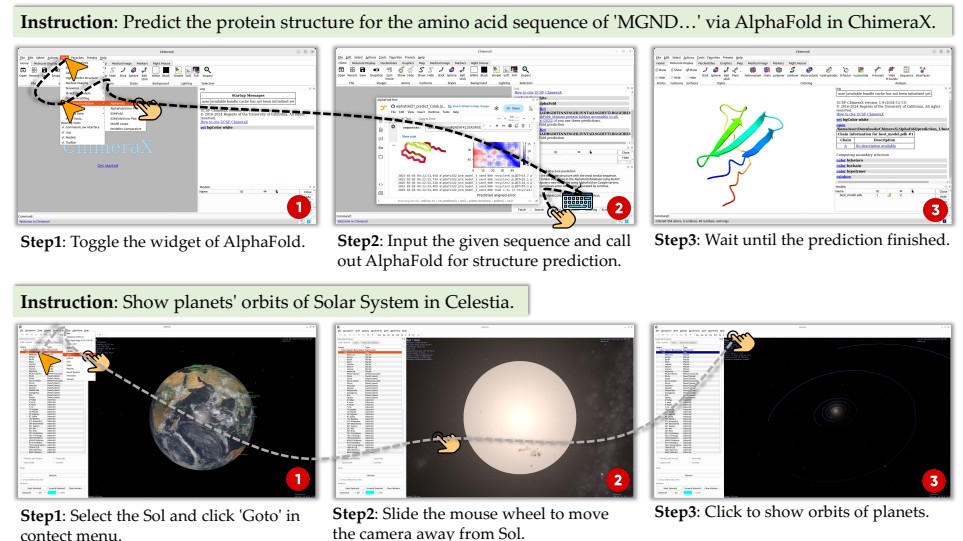

Figure 1: SCIENCEBOARD is a pioneering computer environment for scientific discovery agents, integrated with professional software. It enables agents to autonomously follow instructions and complete realistic scientific tasks by interacting with the system via GUI or CLI.

Recently emerging computer-using agents (Wu et al., 2024; OpenAI, 2025), capable of operating digital devices in a human-like manner, present a promising approach toward achieving these visions. These agents can interact with operating systems through Command-Line Interfaces (CLI; Sun et al., 2024a; Wang et al., 2024d) or perform mouse and keyboard actions via Graphical User Interfaces (GUI; Cheng et al., 2024; Wu et al., 2025b), mimicking the user experience to flexibly automate complex workflows (Xie et al., 2024; Rawles et al., 2025; Hu et al., 2024). As illustrated in Figure 1, to predict the protein structure of an amino acid sequence, the agent launches ChimeraX, selects the AlphaFold widget, and inputs the sequence for prediction. In this way, scientific tasks could be performed through step-by-step autonomous interaction with software.

To initiate the use of computer-using agents to assist human scientists with daily tasks, we introduce SCIENCEBOARD, a novel realistic environment designed for developing AI-powered research assistants. Our infrastructure comprises a scalable framework for scientific exploration that integrates: (1) a flexible ecosystem comprising scientific software across multiple domains, and (2) standardized evaluation pipelines for rigorous assessment. It supports dual-mode interaction, allowing LLM/VLM-based computer agents to operate through either CLI or GUI.

Building upon SCIENCEBOARD, we curate a benchmark comprising 169 tasks that encompass scientific experiment workflows drawn from six scientific domains, including algebra, biochemistry, theorem proving, geographic information systems, astronomy, and scientific documentation. These high-quality and challenging tasks are meticulously designed by annotators with disciplinary backgrounds, simulating the daily routines faced by human scientists. Task completion requires agents to interact with the system via CLI and GUI, exercising a wide range of capabilities—including visual and textual reasoning, tool manipulation, coding, mathematics, spatial understanding, and deep domain-specific knowledge. Unlike widely used desktop applications, scientific software exhibits considerable complexity in I/O formats. Consequently, we reconfigure all software involved to ensure the accuracy and reliability of execution-based evaluation. We design a suite of evaluation functions that verify task completion by retrieving the internal states of the system.

We evaluate widely used LLMs and VLMs as agents on SCIENCEBOARD, incorporating both proprietary models and their open-source counterparts. Across different observation settings, the average success rate of agents ranges between 0% to 15%, with performance peaking at 20% in the most favorable subcategories. This demonstrates that current computer-using agents, while promising, remain far from capable of serving as scientific assistants, largely due to their limited action capability and domain knowledge. Our analysis further reveals their inherent limitations and explores design principles for developing more agents for science.

## 2 RELATED WORKS

**Computer-Using Agents.** Language agents (Sumers et al., 2024) have recently garnered significant attention due to their interactive capabilities (Li et al., 2023; Sun et al., 2024c; Hong et al., 2024; Liu et al., 2024a). Recent studies indicate their potential to interact with operating systems and automate computer tasks as humans do, leading to the proliferation of computer-using agents (OpenAI, 2025). One line of research utilizes Command Line Interface (CLI), where agents generate executable scripts (*e.g.*, Python or Shell scripts) to interact with systems programmatically (Wang et al., 2024b). In this process, agents perform code synthesis (Sun et al., 2024a) or invoke APIs (Wu et al., 2024; Zhang et al., 2024). Another line of research focuses on Graphical User Interface (GUI) agents (Cheng et al., 2024; Wu et al., 2025b; Lin et al., 2024) that interact with digital devices through human-like mouse and keyboard actions (Niu et al., 2024; Zheng et al., 2024; Gou et al., 2025). These agents transform user instructions into executable actions within the operating system (*e.g.*, clicking an icon or scrolling through a page). Powered by VLMs, GUI agents have been applied to automate desktop (Xie et al., 2024) and mobile (Rawles et al., 2025) tasks, as well as specialized engineering workflows (Cao et al., 2024), showing promising paths toward digital automation. This work innovatively initiates the use of computer agents in scientific workflows, taking a step closer to autonomous research assistants.

**AI for Scientific Discovery.** The rapid advancement of LLMs has reshaped the landscape of scientific discovery (Microsoft, 2023), boosting multiple stages of the research cycle (Luo et al., 2025). With the rise of LLM/VLM-based agents, there is a growing demand for these game-changers with college-level knowledge (Wang et al., 2024a) to transcend traditional tasks like question answering (Lu et al., 2022; Krithara et al., 2023; Lu et al., 2024b). Recent efforts have been directed towards harnessing such power to assist with diverse components of the research cycle, including idea and hypothesis generation (Si et al., 2024; Liu et al., 2024b), data analysis (Chen et al., 2025; Gu et al., 2024; Majumder et al., 2024), scientific programming (Tian et al., 2024; Novikov et al., 2025), paper writing (Wang et al., 2024c), and peer-reviewing (Yu et al., 2024). Meanwhile, incorporating domain knowledge or even constructing foundation models (Microsoft, 2025) can endow these agents with the capability to solve domain-specific problems, such as theorem proving (Song et al., 2025), chemical reasoning (Ouyang et al., 2024; Tang et al., 2025) and biological discovery (Wang et al., 2025; Zhao et al., 2025; Wang et al., 2025; Frey et al., 2025). With the vision of constructing autonomous research assistants (Schmidgall et al., 2025), our work represents the first to support agents in executing end-to-end scientific exploration workflows, thereby laying a cornerstone for advancing AI-powered scientific discovery.

## 3 SCIENCEBOARD ENVIRONMENT

In this part, we introduce SCIENCEBOARD environment, which encompasses real-world science software that could be manipulated through GUI and CLI interfaces. The interface is developed based on an Ubuntu virtual machine (VM), serving as the underlying infrastructure. The dynamic and visually intensive environments distinguish SCIENCEBOARD from all previous works that evaluate the scientific capabilities of models or agents.

### 3.1 PRELIMINARIES AND TASK DEFINITION

A computer-using agent receives task instructions, selects actions to manipulate software, and receives feedback reflecting changes in the environment (tabletop). This interaction is modeled as a Partially Observable Markov Decision Process (POMDP), defined by the tuple $\langle g, \mathcal{S}, \mathcal{A}, \mathcal{O}, \mathcal{T} \rangle$, where $g$ is the goal, $\mathcal{S}$ is the state space, $\mathcal{A}$ is the action space, $\mathcal{O}$ is the observation space (including environment feedback), and $\mathcal{T} : \mathcal{S} \times \mathcal{A} \rightarrow \mathcal{S}$ is the state transition function. Given a policy $\pi$, the agent predicts actions at each time step $t$ based on the goal $g$ and memory $m_t = o_j, a_j, o_{j+1}, a_{j+1}, \ldots, o_t$ $(0 \leq j < t)$, which records the sequence of past actions and observations. The trajectory $\tau = [s_0, a_0, s_1, a_1, \ldots, s_t]$ is determined by the policy and environment dynamics:

$$p_\pi(\tau) = p(s_0) \prod_{t=0}^{T} \pi(a_t|g, s_t, m_t) \mathcal{T}(s_{t+1}|s_t, a_t) \tag{1}$$

**Observation and Memory.** We evaluate computer agents using three types of observation spaces: text-only, visual-only, and combined text-visual observations. For text-based observations, we

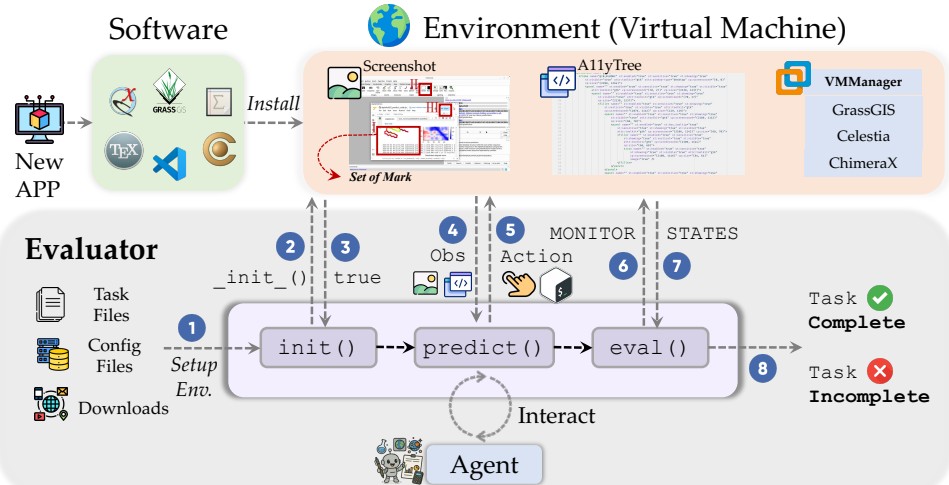

Figure 2: Overview of the SCIENCEBOARD infrastructure. The scalable environment is built upon a VM pre-installed with scientific discovery software. It supports both CLI and GUI interfaces to enable autonomous agent interaction. For each task designed to evaluate the agent's capability as a research assistant, an initialization script, configs, and related files are provided. Agents perceive the environment through visual or textual modalities, and are expected to plan and act accordingly. After the interaction, an evaluation function determines completion based on the VM internal states.

use accessibility trees (`allytree`[1]) to generate structured textual representations of screenshots. For visual observations, we capture high-resolution screenshots directly. The specific observation combinations used in our experiments are detailed in Section 5.1, with further information in Appendix B.5. Our POMDP agent requires memory to retain historical information. Following previous work (Yao et al., 2023; Ma et al., 2024), we construct this memory by concatenating the agent's most recent observations.

**Goal and Unified Action Space.** Each task is specified by a natural language (NL) instruction, such as `Display atoms in sphere style`, describing the user's intended goal. The policy model decomposes a complex goal instruction into a sequence of actions. We specially design a unified action space $\mathcal{A}$ in SCIENCEBOARD, integrating diverse interaction modalities crucial for scientific tasks. For GUI actions, agents can perform the full range of human-computer interactions, including mouse movements, clicks, keystrokes, and other typical input behaviors as in prior work (Xie et al., 2024; Zhou et al., 2024) (*e.g.*, `CLICK[991, 019]`). For CLI actions, agents can interact at two levels: (a) invoking system-level commands within the Ubuntu terminal, and (b) utilizing application-specific CLI or scripting mechanisms. Moreover, $\mathcal{A}$ comprises an `answer` action, enabling agents to provide specific answers for QA tasks, and a `call_api` action, allowing agents to leverage predefined external APIs to broaden their capabilities. A comprehensive list of supported action types is available in Appendix B.4.

**LLM/VLM-based Policy Model.** An LLM / VLM model acts as the policy model to drive the agent's behavior. The policy model receives the current observation and generates the next action accordingly. For pure-text observation, we adopt LLMs as the policy. Otherwise, we leverage VLMs.

### 3.2 SCIENTIFIC DISCOVERY EVALUATION FRAMEWORK

Unlike prior work that primarily focuses on static QA, coding, or single-step tasks, we aim to provide agents with a realistic and visually grounded environment to support autonomous exploration, which in turn introduces greater challenges for planning and action. In SCIENCEBOARD environment, as shown in Figure 2, we (1) simulate scenarios where scientific software is used to solve domain-specific problems, (2) enable agents to interact with the environment through diverse observations, and (3) ensure that agent behaviors can be rigorously evaluated.

---

[1]`allytree`: Accessibility (a11y) trees are hierarchical structures representing UI elements on the screen.

**Scientific Software Installation and Adaptation.** For each domain, we select an open-source application that supports both visual and textual observations as the agent's playground. To enable access to the internal state of each application within the VM, we adapt the software accordingly. Given the complexity and limited completeness of scientific applications, we inject a lightweight server that launches alongside the application's main UI process to expose internal states via HTTP requests. This server is capable of querying the application's runtime internal states, which serve as the basis for downstream evaluation. For applications that do not natively support remote control via RESTful APIs, we modify and recompile their source code to ensure that both UI elements and internal states can be accessed. In addition, the server supports partial state control of the software, allowing us to initialize with specific configurations to simulate contextualized task environments. More about the software selected and further implementation details are provided in Appendix B.3.

**Agent Interactions with the Environment.** The LLM/VLM agent interacts with the environment as described in Section 3.1, receiving observations and executing actions accordingly. Scientific software processes these actions and returns updated states. The agent operates autonomously, continuing this loop until it outputs a signal (`DONE` or `FAIL`) or reaches the predefined attempt limit.

Table 1: Typical evaluation cases of SCIENCEBOARD include exact matching, range-based assessment, and numerical tasks with tolerance. We have tailored appropriate evaluation methods for each task. Additional evaluation strategies are detailed in Appendix D.4.

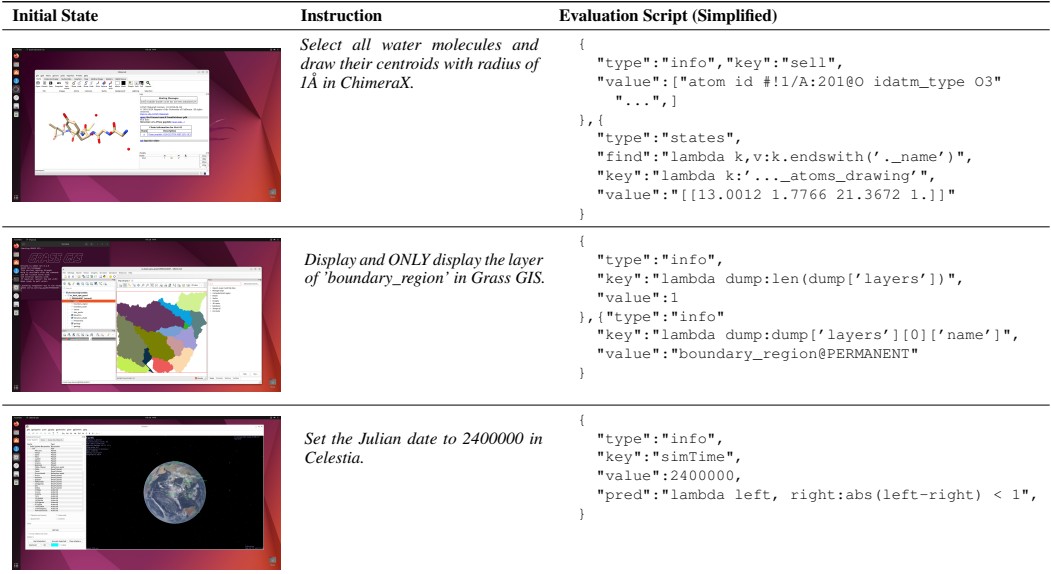

| Initial State | Instruction | Evaluation Script (Simplified) |
|---|---|---|
| | *Select all water molecules and draw their centroids with radius of 1Å in ChimeraX.* | `{`
`  "type":"info","key":"sell",`
`  "value":["atom id #!1/A:201@O idatm_type O3"`
`    "...",]`
`},{`
`  "type":"states",`
`  "find":"lambda k,v:k.endswith('._name')",`
`  "key":"lambda k:'..._atoms_drawing'",`
`  "value":"[[13.0012 1.7766 21.3672 1.]]"`
`}` |
| | *Display and ONLY display the layer of 'boundary_region' in Grass GIS.* | `{`
`  "type":"info",`
`  "key":"lambda dump:len(dump['layers'])",`
`  "value":1`
`},{"type":"info"`
`  "key":"lambda dump:dump['layers'][0]['name']",`
`  "value":"boundary_region@PERMANENT"`
`}` |
| | *Set the Julian date to 2400000 in Celestia.* | `{`
`  "type":"info",`
`  "key":"simTime",`
`  "value":2400000,`
`  "pred":"lambda left, right:abs(left-right) < 1",`
`}` |

**Evaluation Pipeline.** Given the complexity of scientific tasks, conventional answer-matching metrics and even execution-based evaluations (Xie et al., 2024; Zhou et al., 2024), often lack the granularity required to assess workflows accurately. For instance, as shown in Table 1, the rotation of a protein does not affect the correctness of visualization, whereas computational tasks in astronomy are usually influenced by the current clock state. Therefore, we propose a fine-grained evaluation based on both the correctness of key I/O during the workflow and the final state of the VM.

To handle the diverse criteria for determining task correctness (*e.g.*, exact matching, range-based assessment, numerical tolerance, file comparison), we design a set of evaluation templates. For each specific task, the relevant template is then instantiated with the appropriate parameters and expected gold standard values. This ensures both consistent validation and scalability for future extension. More evaluation details are in Appendix B.2.

## 4 SCIENCEBOARD BENCHMARK

In this section, we present the covered domains, the annotation pipeline, and statistics of the benchmark constructed based on the SCIENCEBOARD environment.

## 4.1 DOMAIN AND TASK COVERAGE

As a pioneering benchmark for scientific exploration, SCIENCEBOARD spans six domains selected for their relevance to key stages of the scientific workflow, such as simulation, modeling, prediction, and knowledge (Microsoft, 2023). In selecting software for each domain, we consider not only its representativeness, but also practical criteria for evaluation: open-source availability, `a11ytree` compatibility, and no requirement for user authentication.

(1) **Biochemistry.** We employ `UCSF ChimeraX` (Goddard et al., 2018; Meng et al., 2023), a molecular analysis tool that supports structural modeling (*e.g.*, AlphaFold (Jumper et al., 2021)). The tasks assess the agent's ability to manipulate biomolecular structures, as well as to reason over spatial conformations and biochem annotations.

(2) **Algebra.** `KAlgebra` is employed to evaluate the agent's potential in symbolic mathematics. Tasks involve executing algebraic expressions, interpreting plots, and manipulating symbolic functions. These scenarios require the agent to exhibit strong mathematical symbolic reasoning and visual grounding capability.

(3) **Theorem Proving.** We use `Lean 4` (Moura & Ullrich, 2021) as a proof assistant to assess agents' abilities in formal logic and deductive reasoning. The ATP tasks in this category emphasize syntactic precision and logical coherence, evaluating the agent's capability to generate semantically valid formal proofs.

(4) **Geographic Information System.** `GrassGIS`, a computational engine for raster, vector, and geospatial processing, is included to examine the agent's skills in understanding terrain, hydrology, and handling spatio-temporal data, with support for functions such as ecosystem modeling.

(5) **Astronomy.** We integrate `Celestia`, a planetarium software simulating real-world astronomical scenarios. Agents must demonstrate temporal-spatial awareness and knowledge of the cosmos and celestial objects by tracking planetary systems, simulating orbital events, and querying object metadata across time and space.

(6) **Scientific Documentation.** To simulate research documentation workflows, we adapt and incorporate `TeXstudio` to assess the agent's technical writing capabilities. In standalone tasks, agents are expected to compose well-structured abstracts, generate plots, and produce formal reports based on provided instructions. In cross-application scenarios, `TeXstudio` is coupled with the aforementioned software to evaluate whether agents can extract meaningful insights from experiments and synthesize them into coherent narratives.

These domains enable evaluating a science agent's capabilities across multiple dimensions, including visual / textual reasoning, math, coding, tool use, spatial understanding, domain-specific knowledge, and more. Additionally, to explore the potential for end-to-end scientific automation, documentation tasks are integrated with other domains to support cross-application workflows—such as automatically generating an experimental report based on completed upstream tasks. More details about the software platforms used to instantiate and convey the tasks in SCIENCEBOARD are provided in Appendix B.3.

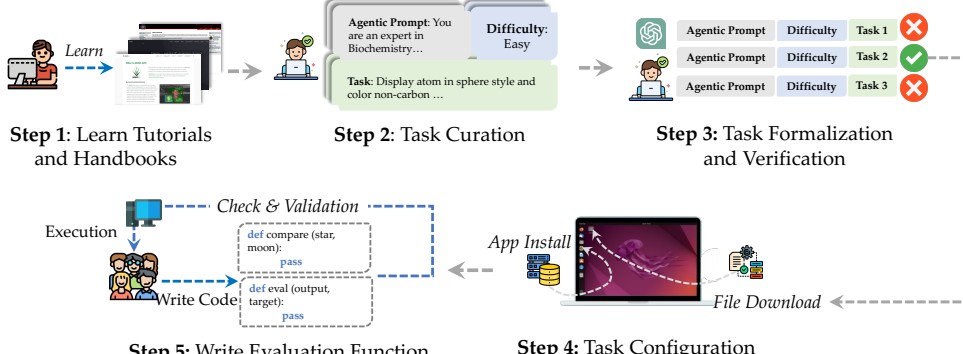

Figure 3: The annotation pipeline of the tasks in SCIENCEBOARD benchmark.

## 4.2 Task Annotation Pipeline

To effectively construct tasks that are appropriately challenging, diverse, and aligned with the features of scientific software, we leverage a pipeline that spans from training annotators with tutorials and handbooks to conducting execution-based validation, as shown in Figure 3.

(1) **Tutorial Learning.** Five annotators initially collect and learn from tutorials and handbooks related to the software. After that, each annotator studies and explores a software's basic unit operations, *e.g.*, plotting the Bernoulli lemniscate in KAlgebra. Details are in Appendix D.1.

(2) **Task Curation.** Each annotator selects a scientific software, installs it within SCIENCEBOARD, and begins drafting task instructions based on its functionalities. Task types include but are not limited to: configuration, simulation, QA, and domain-specific expertise. Each task is tentatively assigned a difficulty. Thereafter, agentic prompts aligned with the drafted tasks will be curated.

(3) **Formalization and Selection.** Different annotators exhibit varying linguistic habits, we employ ChatGPT to standardize the task format. Annotators then conduct a cross-check, excluding those lacking diversity, poor executability, or non-unique answers, to finalize the set of tasks for use.

(4) **Configuration Function Writing.** The purpose of this step is to initialize the software and provide specific contexts, *e.g.*, supplying a map for GIS tasks or a protein sequence for biochemistry tasks. Annotators will write a set of functions for each software to modify the VM status, *i.e.*, the internal state of the software, along with general configuration functions (*e.g.*, downloading required files). Tasks commence only after all initialization have been successfully executed.

(5) **Evaluation Function Writing and Validation.** Evaluation functions are developed to assess task outcomes rigorously. As described in Section 3.2, evaluations are state-based, with functions derived from a base evaluator template. Annotators retrieve the task state from the VM and assess it based on criteria such as I/O matching and predefined ranges. The function returns either "task complete" or "task fail." Cross-validation is performed for consistency, with each task executed by two randomly selected annotators on separate VMs. The results are analyzed to ensure the evaluator's correctness, even under intentional attempts by annotators to deceive the system.

## 4.3 Task Statistics

The task statistics of SCIENCEBOARD benchmark are presented in Table 2. Specifically, it comprises 169 unique tasks across 6 domains, with task difficulty categorized into three levels. We curate a balanced number of tasks that are representative enough while keeping the evaluation cost manageable. During annotation, we define multiple task types to evaluate agents' ability to perform diverse operation flows and leverage domain-specific knowledge.

Table 2: Statistics of SCIENCEBOARD.

| Task Type | Statistics |
|---|---|
| **Total Tasks** | **169 (100%)** |
| - GUI | 38 (22.5%) |
| - CLI | 33 (19.5%) |
| - GUI + CLI | 98 (58.0%) |
| **Difficulty** | |
| - Easy | 91 (53.8%) |
| - Medium | 48 (28.4%) |
| - Hard | 28 (16.6%) |
| - Open Problems | 2 (1.2%) |
| **Instructions** | |
| Avg. Length of Task Instructions | 20.0 |
| Avg. Length of Agentic Prompt | 374.9 |
| **Execution** | |
| Avg. Steps | 9.0 |
| Avg. Time Consumption | 124(s) |

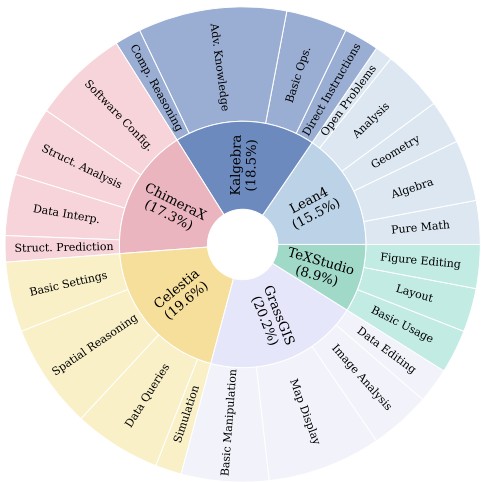

Figure 4: Distribution of tasks in SCIENCE-BOARD benchmark.

The distribution of task types is shown in Figure 4. Beyond the innovation of a realistic environment, SCIENCEBOARD benchmark also improves upon prior work in terms of task design and content

diversity. More details about task diversity, stability analysis, and comparison with representative scientific benchmarks are provided in Appendix D.

# 5 EXPERIMENTS

## 5.1 EXPERIMENTAL SETTINGS

**Backbones.** We employ three types of backbones for agents. These include **proprietary models**: `GPT-4o` (Hurst et al., 2024), `GPT-5`/ `GPT-5.5` (OpenAI, 2025b), `Claude-3.7-Sonnet` (Anthropic AI, 2024), `Claude-Opus-4.6` (Anthropic, 2025), `Gemini-2.0-Flash` (Gemini Team, 2024), `Gemini-2.5-Pro` (Gemini Team, 2025) and `o3-mini` (OpenAI, 2025a); **open-source models**: Qwen2.5-VL-72B-Instruct (Bai et al., 2025), InternVL3-78B (Chen et al., 2024), QvQ-72B-Preview (Qwen Team, 2024), and GPT-oss-120B (Open AI, 2025); and **GUI action models**: OS-Atlas-Pro-7B (Wu et al., 2025b), UGround-V1-7B (Gou et al., 2025), UI-TARS-72B-DPO / UI-TARS-1.5-7B (Qin et al., 2025), and GUI-Actor-7B (Wu et al., 2025a). More details in Appendix E.1.

**Observation Space.** We follow established observation settings (Xie et al., 2024; Zhou et al., 2024): (1) full desktop screenshots; (2) `allytree`, a structured text-only representation; (3) Screenshots + `allytree`; and (4) Set-of-Marks (Yang et al., 2023), which partitions images into marked regions to aid grounding. Further details are in Appendix B.5.

## 5.2 RESULTS

We compare the performance of computer-use agents powered by different LLMs and VLMs on SCIENCEBOARD, as presented in Table 3. We summarize our key empirical findings as follows:

**Performance Hierarchy.** Existing agents remain far from being capable of effectively assisting human scientists in completing real-world scientific exploration tasks. Even SOTA models, such as `GPT-5` and `Gemini-2.5-Pro`, achieve an average success rate of only 20%. Across various settings, open-source counterparts can partially match proprietary models. However, they still exhibit markedly lower overall performance, with an average success rate of less than 12% and approaching nearly 0% in some task categories. The gap between agent and human performance underscores the limitations of the status quo and necessitates further research.

**Domain-Specific Performance Insights.** Across domains, we observe clear performance imbalances: models perform moderately well on Algebra and Biochemistry but degrade notably on GIS and Astronomy. We attribute this to: (1) Interfaces: Algebra and Biochemistry tasks often support both CLI and GUI execution, whereas GIS and Astronomy rely mainly on GUI interactions. Agents generally handle CLI commands more reliably than fine-grained GUI grounding, which demands precise visual localization. (2) Task emphasis: Geographical and astronomical tasks involve dense visual elements (e.g., maps, star charts), making it difficult for agents to identify and reason over relevant information. This also indicates the limited 3D spatial reasoning ability of current VLMs.

**Impact of Different Observations.** Different observation modalities have a significant impact. Overall, `allytree` + screenshots setting yields the best performance. In other settings, Qwen2.5-VL performs exceptionally well under screenshot setting, which we attribute to its advanced GUI ability. Under `allytree`, the attribute information of elements allows LLMs to complete certain tasks by relying solely on textual observations. Meanwhile, we observe that the SoM sometimes introduces negative effects. It is likely that although SoM provides bounding boxes to ease grounding, scientific software often contains massive elements on screen (*e.g.*, dense celestial objects and complex cosmic backgrounds), which introduces substantial noise and increases the difficulty of visual reasoning.

**Impact of Compute Scaling and Native Reasoning.** To determine if test-time compute can mitigate these limitations, we further evaluate frontier models with extended generation limits and native reasoning modes in Appendix F.4. Our findings indicate that scaling inference-time compute yields tangible but bounded improvements. For instance, allocating high reasoning effort increases success rates in the Biochemistry and GIS domains. However, this scaling does not fundamentally

Table 3: Success rates on SCIENCEBOARD. We present the performance of each agent backbone across different scientific domains under various observation settings. Proprietary Models , Open-Source VLMs / LLMs , and GUI Action Model are distinguished by color.

| Obs. | Model | Success Rate (↑) | | | | | | |
|---|---|---|---|---|---|---|---|---|
| | | Algebra | Biochem | GIS | ATP | Astron | Doc | Overall |
| Screenshot | GPT-5.5 | 9.68% | 41.38% | 11.76% | 3.85% | 18.18% | 25.00% | 18.31% |
| | GPT-5 | 6.45% | 24.14% | 0.00% | 0.00% | 12.12% | 12.50% | 9.20% |
| | GPT-4o | 3.23% | 0.00% | 0.00% | 0.00% | 0.00% | 6.25% | 1.58% |
| | Claude-Opus-4.6 | 3.23% | 68.97% | 2.94% | 0.00% | 6.06% | 6.25% | 14.58% |
| | Claude-3.7-Sonnet | 9.67% | 37.93% | 2.94% | 0.00% | 6.06% | 6.25% | 10.48% |
| | Gemini-2.5-Pro | 6.45% | 31.03% | 0.00% | 0.00% | 0.00% | 12.50% | 8.33% |
| | Gemini-2.0-Flash | 6.45% | 3.45% | 2.94% | 0.00% | 0.00% | 6.06% | 3.15% |
| | InternVL3-78B | 6.45% | 3.45% | 0.00% | 0.00% | 0.00% | 6.25% | 5.46% |
| | UI-TARS-1.5-7B | 12.90% | 13.79% | 0.00% | 0.00% | 6.06% | 0.00% | 2.69% |
| | Qwen3-VL-235B | 9.68% | 27.59% | 0.00% | 0.00% | 9.09% | 12.50% | 9.81% |
| | Kimi-K2.5 | 9.68% | 27.59% | 0.00% | 0.00% | 3.03% | 6.25% | 7.76% |
| a11ytree | GPT-4o | 12.90% | 20.69% | 2.94% | 0.00% | 6.06% | 0.00% | 7.10% |
| | Claude-3.7-Sonnet | 19.35% | 34.48% | 2.94% | 3.85% | 12.12% | 0.00% | 12.12% |
| | Gemini-2.0-Flash | 9.68% | 17.24% | 0.00% | 0.00% | 0.00% | 0.00% | 4.49% |
| | o3-mini | 16.13% | 20.69% | 2.94% | 3.85% | 15.15% | 6.25% | 10.84% |
| | Qwen2.5-VL-72B | 9.68% | 10.34% | 2.94% | 0.00% | 3.03% | 0.00% | 4.33% |
| | InternVL3-78B | 3.23% | 3.45% | 0.00% | 0.00% | 0.00% | 0.00% | 1.11% |
| | GPT-oss-120B | 19.35% | 13.79% | 0.00% | 0.00% | 9.09% | 0.00% | 7.04% |
| Screenshot + a11ytree | GPT-5 | 41.93% | 62.07% | 5.88% | 7.69% | 15.15% | 12.50% | 24.20% |
| | GPT-4o | 22.58% | 37.93% | 2.94% | 7.69% | 3.03% | 12.50% | 14.45% |
| | Claude-3.7-Sonnet | 12.90% | 41.37% | 8.82% | 3.85% | 9.09% | 18.75% | 15.79% |
| | Gemini-2.5-Pro | 16.13% | 55.17% | 2.94% | 0.00% | 15.15% | 12.50% | 16.98% |
| | Gemini-2.0-Flash | 16.13% | 24.14% | 2.94% | 0.00% | 18.18% | 12.50% | 12.32% |
| | Qwen2.5-VL-72B | 16.13% | 20.69% | 2.94% | 0.00% | 18.18% | 12.50% | 11.74% |
| | InternVL3-78B | 6.45% | 3.45% | 0.00% | 0.00% | 3.03% | 6.25% | 3.20% |
| Set-of-Mark | GPT-4o | 6.45% | 3.45% | 0.00% | 0.00% | 3.03% | 12.50% | 4.24% |
| | Claude-3.7-Sonnet | 16.13% | 31.03% | 5.88% | 0.00% | 6.06% | 12.50% | 11.93% |
| | Gemini-2.0-Flash | 3.23% | 0.00% | 0.00% | 0.00% | 3.03% | 6.25% | 2.09% |
| | Qwen2.5-VL-72B | 6.45% | 6.90% | 2.94% | 0.00% | 3.03% | 12.50% | 6.36% |
| | QvQ-72B-Preview | 0.00% | 0.00% | 2.94% | 0.00% | 3.03% | 0.00% | 0.49% |
| | InternVL3-78B | 3.23% | 6.90% | 2.94% | 0.00% | 0.00% | 0.00% | 2.18% |
| Human Performance | | 74.19% | 68.97% | 55.88% | 42.31% | 51.52% | 68.75% | 60.27% |

resolve the overall performance deficit. This suggests that the primary bottleneck for current agents in scientific workflows is not merely the depth of cognitive reasoning, but rather general computer-using and agentic capabilities, which require accurately perceiving dense, domain-specific UI elements and translating high-level scientific plans into precise, executable actions.

**From Computer-using Agents to AI Co-scientist.** Agent backbones excelling on general computer-using benchmarks such as OSWorld (Xie et al., 2024) do not straightforwardly dominate scientific tasks, despite the largely shared action space. This gap indicates that building agents for science requires a capability axis beyond planning and action: agents additionally need domain knowledge, spanning both scientific reasoning (*e.g.*, interpreting a protein's spatial conformation or a hydrological raster) and software-specific expertise (*e.g.*, knowing the right module in Grass-GIS). Also, it requires flexible GUI + CLI hybrid execution, since some scientific software exposes overlapping but non-identical functionality across the two modalities, with CLI frequently offering a

more concise path for complex operations, provided the agent knows how to use it. Choosing which interface to leverage is a learned skill largely absent from GUI-only capabilities.

# 6 ANALYSIS

To further investigate the factors influencing agents' capabilities, we conduct additional analysis to understand the underlying causes and the behavioral differences among heterogeneous models.

**GUI vs. Hybrid.** Some tasks support both GUI and CLI as interchangeable interfaces. For example, ChimeraX offers nearly full functional coverage through both modes for biochemistry tasks. To test how computer-using agents handle such hybrid software, we disable ChimeraX's CLI, enforcing GUI-only execution (`allytree` + screenshot). As shown in Figure 5, `GPT-4o` and InternVL3 suffer clear drops in performance, whereas Qwen2.5-VL remains largely unaffected, indicating better adaptation to GUI execution.

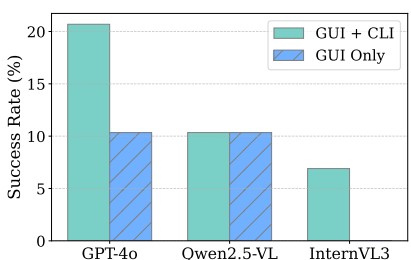

Figure 5: GUI + CLI v.s. GUI Only.

These results suggest that future agents should be more adaptable and equipped with stronger GUI capabilities to remain robust across hybrid and vision-only settings. Extended analyses are provided in Appendix F.

**Disentangled Planning and Action.** Observations from failure cases indicate that some models, such as `GPT-4o`, can effectively plan tasks but lack sufficient grounding capabilities. Therefore, we explore separating planning and action. Following existing practices (Wu et al., 2025b), we configure `GPT-4o` as the planner and utilize various VLMs and GUI action models as the grounding models.

Table 4: Success rates of different VLM agent combinations under the planner + grounding model setting on SCIENCEBOARD. The observation setting used in this experiment is screenshot. Colors denote Proprietary Models , Open-Source VLMs and GUI Action Models.

| Planner | Grounding Model | Success Rate (↑) | | | | |
|---|---|---|---|---|---|---|
| | | Algebra | Biochem | GIS | Astron | Overall |
| GPT-4o | OS-Atlas-Pro-7B | 6.25% | 10.34% | 0.00% | 3.03% | 4.92% |
| | UGround-V1-7B | 0.00% | 3.45% | 0.00% | 3.03% | 1.62% |
| | Qwen2.5-VL-72B | 12.50% | 34.48% | 11.76% | 9.09% | 16.96% |
| | UI-TARS-72B | 3.23% | 10.34% | 5.88% | 6.06% | 6.38% |
| | GUI-Actor-7B | 21.88% | 44.83% | 2.94% | 12.12% | 20.44% |
| GPT-4o | | 3.23% | 0.00% | 0.00% | 0.00% | 0.81% |

The results in Table 4 show that modular approaches yield significant improvements and are promising for tackling complex and visually demanding tasks in scientific software workflows.

# 7 CONCLUSION

We propose SCIENCEBOARD, a first-of-its-kind realistic environment designed to empower autonomous agents in scientific exploration with rigorous validation. Building upon our infrastructure, we curate a highly challenging benchmark of diverse scientific tasks meticulously crafted by human experts. Through extensive experiments and analysis, we found that even state-of-the-art computer-using agents perform significantly below human-level proficiency. Although the realization of autonomous agents for scientific discovery remains a distant goal, this work offers actionable insights for future development, and we believe it constitutes advancing AI-powered scientific discovery.

ETHICS STATEMENT

Computer-using agents operating in live OS environments could potentially affect the normal functioning of the system. This is non-negligible in scientific workflows, where a poorly controlled agent could potentially misconfigure experiments, corrupt sensitive research data, or even lead to irreversible data loss. However, considering that all settings in this work are conducted within isolated virtual environments, we do not view this as a concern.

ACKNOWLEDGEMENT

We would like to express our sincere gratitude to the anonymous reviewers from ICLR and the WCUA@ICML workshop for their insightful comments and valuable suggestions, which have significantly helped improve the quality of this work. This research is supported by the WYNG Foundation (HKU 25AG100407). We gratefully acknowledge their support.

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

## LARGE LANGUAGE MODEL USAGE

In this submission, we employed LLMs to aid and polish writing, including grammar and typo checking, as well as for identifying related works.

## LIMITATIONS AND BROADER IMPACTS

As a pioneering effort marking the early stages of integrating computer-using agents into scientific workflows, it is important to acknowledge certain limitations. While our current evaluation, based on both VM states and key I/O correctness, provides robust validation, its reliance on a binary success flag may not fully capture process correctness or partial task completion (*e.g.*, an agent succeeding in most steps but failing at a final one). Introducing a "partial credit" could offer more granular evaluation, but accurately defining and implementing such a system for open-ended, OS-level tasks within diverse scientific software presents significant challenges due to vast state / action spaces. One potential direction for improvement is to introduce VLMs to serve as judges capable of assigning partial credit and providing richer feedback. We leave this as future work.

# A  DISCUSSION AND FUTURE DIRECTIONS

SCIENCEBOARD represents a significant advance in using autonomous agents for scientific workflows. Our findings suggest several key directions for future research:

**Harmonized Domain Knowledge and Agentic Capability.**   Our evaluations suggest that one contributing factor to current agents' limitations in scientific exploration is their insufficient domain knowledge. For instance, the GUI action models we evaluated, while effective at automation, lack the specialized understanding required for complex scientific tasks. Therefore, future advancements may focus on enhancing domain-specific abilities, such as enhancing scientific comprehension (Li et al., 2024), learning from highly relevant resources such as manuals and tutorials, and enabling on-demand knowledge retrieval (Lála et al., 2024). A key challenge will be to effectively harmonize this specialized knowledge with general agentic capabilities (Xu et al., 2024a).

**Collaborative and Specialized Agents as a Solution.**   Analysis in Table 4 indicates that even a basic modular approach of separating planning and action to different agents can yield significant performance improvements in complex scientific software workflows. This points toward developing sophisticated multi-agent systems composed of specialized, heterogeneous agents (Jia et al., 2024a; Ghafarollahi & Buehler, 2024; Agashe et al., 2025). For example, responsibilities could be disentangled by assigning planning to agents capable of deep reasoning (Li et al., 2025), action execution to specialized GUI action models (Wu et al., 2025b; Xu et al., 2024b), and domain-specific capability to models in particular disciplines (Microsoft, 2023; 2025). These agents could be plug-and-play, allowing flexible application across broader aspects of the scientific lifecycle, such as data analysis (Chen et al., 2025), scientific plotting (Jia et al., 2024b), and paper revision (Yu et al., 2024). While promising, it also demands more sophisticated multi-agent designs to manage and coordinate the intricate and multifaceted nature of scientific tasks.

**Extending Digital Agents to Physical Laboratory.**   Current AI-assisted scientific workflows are primarily at the digital level, focusing on tasks such as data analysis, simulation, and software control. A natural and impactful next step is to extend the capabilities of such autonomous agents, as fostered and benchmarked in SCIENCEBOARD, into physical laboratory environments. This transition involves interfacing agents with robotic systems (Burger et al., 2020; Angelopoulos et al., 2024), applying principles of embodied AI to perceive and interact with the physical world. Agents would manipulate laboratory instruments and samples, carry out experimental protocols, and monitor physical processes in real time, thereby fostering a "lab-in-the-loop" (Frey et al., 2025) future where experimentation and AI-driven methods are mutually reinforcing.

# B  DETAILS OF SCIENCEBOARD ENVIRONMENT

## B.1  ENVIRONMENT SETUP

Virtual machines can operate their own kernel and system, enabling compatibility with a wide variety of operating systems. For experiments covered in this paper, we utilize a Linux environment (Ubuntu 22.04.1 LTS with kernel 6.8.0-57-generic) running on x64 personal computers.

## B.2  EVALUATION CRITERIA

As stated in Section 3.2, we employ a fine-grained evaluation methodology based on:

- The final state of the VM (Determinant)
- I/O states and intermediate steps (Non-Determinant)

While the final state of the VM often provides a determinant measure of overall task completion, the diverse nature of I/O and intermediate steps necessitates a varied set of criteria. The following outlines the primary principles applied for I/O correctness:

- **Exact Match:**
    - Strict equality: The output or relevant state must be exactly identical to the gold standard (e.g., for specific textual outputs or numerical values).

- Set equality of lines: For multi-line textual outputs, the content of all lines must match the gold standard, but their order may not be strictly enforced.
- Question-answering: The agent's provided answer to a question is compared against a correct answer or set of acceptable answers.

- **Predicate Satisfaction:** Verifying if specific information and generated outputs satisfy predefined logical conditions or predicates. This includes:

  - Value Existence: A required value, file, or UI element is present as expected.
  - Value Non-Existence: A specified value, file, or UI element is correctly absent.
  - Range Check: A numerical output or parameter falls within a predefined acceptable range (often with a specified tolerance).

- **Correct Task Failure (FAIL):** The agent correctly identifies a task as infeasible or terminates appropriately when unable to complete the objective, outputting a designated `FAIL` signal.

- **Domain-Specific Success Markers:** For certain domains, unique success criteria are employed:

  - Lean Tasks: Successful compilation of the generated Lean proof code is considered a primary indicator.

## B.3 SELECTION AND MODIFICATION OF SCIENTIFIC SOFTWARE

To ensure both technical feasibility and representative task diversity, we selected software tools based on the following criteria:

1. **Accessibility.** The software must be open-source or freely available, allowing transparent integration and reproducibility of experiments.
2. **GUI Compatibility.** The software must expose a usable accessibility tree (a11y tree) to support fine-grained GUI grounding and interaction.
3. **Domain Representativeness.** The software should be representative of key scientific and technical domains, enabling meaningful assessment of multimodal agent capabilities across different types of tasks.

Based on these principles, we selected the following software for each target domain:

- **Lean.** A functional programming language and interactive theorem prover grounded in dependent type theory (specifically Martin-Löf Type Theory). Lean enables formal verification of mathematical theorems and software correctness through rigorous type checking and logical inference, supporting robust development of maintainable and accurate code.

- **ChimeraX.** A next-generation molecular visualization software developed by UCSF, designed for detailed interactive exploration, visualization, and analysis of protein and biomolecular structures. ChimeraX enhances performance and user experience compared to its predecessor, UCSF Chimera, offering improved graphics rendering, extensibility via plugins, and streamlined workflows for structural biology research.

- **KAlgebra.** An educational calculator and graphical plotting application within the KDE Education Project. It supports a wide range of numerical, logical, symbolic, and analytical computations, enabling users to visualize mathematical functions interactively in both two-dimensional (2D) and three-dimensional (3D) environments, thus effectively bridging computational mathematics and educational usability.

- **Celestia.** A cross-platform, interactive real-time 3D astronomical simulation software that allows users to explore the universe through detailed, dynamic visualizations. Celestia is highly extensible via scripting, empowering educational and professional users to model and visualize celestial phenomena and space missions with precision and customization.

- **GrassGIS.** An advanced Geographic Information System (GIS) supporting both raster and vector geospatial data, along with powerful analytical capabilities for spatial modeling, hydrological analysis, and environmental simulations. GrassGIS includes a comprehensive Python API for automation and custom analysis, enabling complex geospatial and temporal analyses tailored to diverse research and application scenarios.

- **TeXstudio.** An integrated LATEX editor that provides a writing environment tailored specifically for creating and managing complex technical and scientific documents. TeXstudio enhances productivity through features such as syntax highlighting, real-time document preview, automatic reference checking, and intuitive assistance tools, greatly simplifying the process of technical writing and document preparation.

## B.4 DETAILS OF ACTION SPACE

The action space employed in SCIENCEBOARD is shown in Table 5. We combine standard interaction primitives (such as GUI operations) with the flexibility of system-level and application-specific Command-Line Interfaces (CLIs), and has been further expanded with several augmented actions tailored for scientific workflows.

Table 5: Action space of SCIENCEBOARD environment.

| Action | Description |
|---|---|
| moveTo(x, y) | Moves the mouse to the target coordinate. |
| moveRel(x, y) | Moves the mouse by an offset from current position. |
| dragTo(x, y) | Drags the mouse to the target coordinate. |
| dragRel(x, y) | Drags the mouse by an offset from current position. |
| click(x, y) | Clicks at the target coordinate. |
| rightClick(x, y) | Performs a right click at the target coordinate. |
| middleClick(x, y) | Performs a middle click at the target coordinate. |
| doubleClick(x, y) | Performs double clicks at the target coordinate. |
| tripleClick(x, y) | Performs triple clicks at the target coordinate. |
| mouseDown(x, y, button) | Presses a mouse button down. |
| mouseUp(x, y, button) | Releases a mouse button up. |
| DONE | Agent decides the task is finished. |
| FAIL | Agent decides the task is infeasible. |
| WAIT [n] | Agent decides it should wait, 'n' defaults to 5(s). |
| ANS [s] | Agent decides it should submit an answer, 's' denotes the answer. |
| API [name, args] | Invokes a registered API call with name and arguments. |
| CODE | Run a generated code script (for in-app / system-level tasks, or custom functions). |

## B.5 DETAILS OF OBSERVATION SPACE

We primarily adhere to well-established settings (Xie et al., 2024; Zhou et al., 2024) for observation space, encompassing: (1) Screenshots, which consist of a full desktop screenshot as observed by human users; (2) a11ytree, a structured text-only representation without visual information, applicable for agents that take pure text input; (3) Screenshots + a11ytree, a hybrid approach that combines and complements both textual and visual modalities; and (4) Set-of-Marks (Yang et al., 2023), a visual prompting method aimed at enhancing the visual grounding capabilities by partitioning an image into marked regions. Details are as follows:

**Screenshot.** We capture a screenshot of the entire computer screen. For screen resolution, we set a default value of 1920×1080, and it also offers a 16:9 aspect ratio. Following OSWorld (Xie et al., 2024), our environment also supports modifying the resolution of virtual machines to avoid potential memorization of absolute pixel values and to assist studies on topics like generalization across different resolutions.

**A11ytree.** An a11ytree refers to an intricate structure generated by the browser or OS accessibility APIs that renders a representative model of the content, providing a means of interaction for assistive technologies. Each node within the accessibility tree hosts important information about a UI element. In SCIENCEBOARD, which utilizes an Ubuntu-based GNOME desktop environment, we employ the Assistive Technology Service Provider Interface [2]. Specifically, we adopt pyatspi to programmatically retrieve the accessibility tree on Ubuntu.

To make complex a11ytree tractable, and critically, to ensure they fit within the context length of open-source models, we filter out non-essential elements. This filtering is performed based on

---

[2] https://docs.gtk.org/atspi2/

element attributes such as their tag, visibility, and availability. For the elements that remain after filtering, only key information—specifically their tag, name, text, position, and size—is retained and subsequently concatenated to form the input representation for the agent.

**Screenshot + a11ytree.** To further enhance the action execution capabilities of computer-using agents, especially for models with weaker grounding abilities, we utilize a combined input of screenshots and `a11ytree`.

**Set-of-Mark.** We follow the official implementation of Set-of-Mark (Yang et al., 2023). We leverage the information from the filtered `a11ytree` and mark the elements on the screenshot with a numbered bounding box. Following VisualWebArena (Koh et al., 2024) and UFO (Zhang et al., 2024), we further combine the annotated screenshot with the text metadata from `a11ytree`.

## C  ACCESSING SCIENCEBOARD ENVIRONMENT

To facilitate broader adoption and reproducibility, we provide several methods for accessing SCIENCEBOARD environment. Researchers can choose the most suitable option based on their technical requirements and resources:

**Direct Deployment.** The entire framework, including all scientific software and evaluation scripts, is available for direct deployment on a native Ubuntu system. Full setup instructions and dependency lists are provided in our repository.

**Docker Container.** We also provide a Docker image that encapsulates the environment, making it easy to run SCIENCEBOARD across different machines and operating systems, which is available at https://anonymous.4open.science/r/ScienceBoard/.

**Cloud Platforms.** For scalability and powerful computational resources, SCIENCEBOARD can be deployed on cloud platforms like Amazon Web Services (AWS). We will provide guidelines upon acceptance.

## D  DETAILS OF SCIENCEBOARD BENCHMARK

### D.1  TASK ANNOTATION

During the task annotation process, we primarily utilize the tutorials and handbooks listed in Table 6 to guide annotators in exploring the relevant domain and corresponding software and tools. All app data collection and task creation are completed by the authors.

### D.2  TASK DIVERSITY

To explore the diversity of tasks in SCIENCEBOARD, we perform a t-SNE (van der Maaten & Hinton, 2008) visualization, as shown in Figure 6. We obtain embeddings for all task instructions using `text-embedding-3-small` and then apply t-SNE to reduce their dimensionality to two for visualization. The semantic distribution of instructions clearly distinguishes tasks across different domains, while also revealing considerable diversity within each individual domain. Furthermore, we can observe some intersections between Scientific Documentation tasks and tasks from other domains, which reflects the presence of cross-application workflows in our benchmark.

### D.3  COMPARISON WITH EXISTING BENCHMARKS

We compare SCIENCEBOARD with existing well-established benchmarks for scientific tasks, as shown in Table 7.

SCIENCEBOARD is the first to offer a realistic environment for evaluating scientific tasks. In terms of I/O, it incorporates structured code input and visual information, which are critical for simulating scientific experiment workflows. It also supports GUI automation, making it well-suited for visual

Table 6: Sources of the tutorials and handbooks employed in the task annotation process.

| Software | Tutorial & Handbook Sources |
|---|---|
| Kalgebra | https://docs.kde.org/stable5/en/kalgebra/kalgebra/index.html |
| ChimeraX | https://www.cgl.ucsf.edu/chimerax/tutorials.html |
| | https://kpwulab.com/wp-content/uploads/2022/04/chimerax-tutorial-kpwulab-2022-0429.pdf |
| Lean 4 | https://lean-lang.org/theorem_proving_in_lean4/ |
| | https://leanprover-community.github.io/mathematics_in_lean/index.html |
| | https://lean-lang.org/doc/reference/latest/ |
| Grass GIS | https://grass.osgeo.org/grass84/manuals/index.html |
| | https://neteler.gitlab.io/grass-gis-analysis/ |
| Celestia | https://celestiaproject.space/guides.html |
| | https://en.wikibooks.org/wiki/Celestia |
| | https://celestiaproject.space/docs/CELScriptingGuide/Cel_Script_Guide_v1_0g.htm |
| TeXStudio | https://texstudio-org.github.io/getting_started.html |
| | https://latex-tutorial.com/tutorials/ |

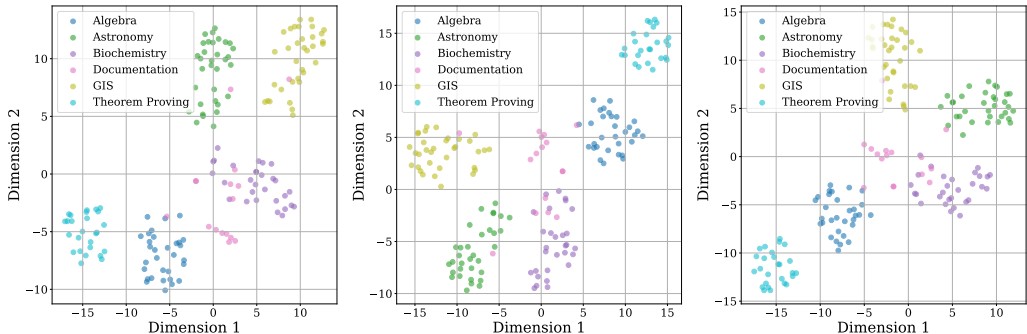

Figure 6: t-SNE visualization of task instructions distribution. The seeds of t-SNE are randomly sampled for each plot.

| Feature | SCIENCEBOARD (our work) | ScienceQA (Lu et al., 2022) | SciCode (Tian et al., 2024) | ScienceAgentBench (Chen et al., 2025) |
|---|---|---|---|---|
| *I/O Formats* | | | | |
| Code / Structured Input | ✓ | ✗ | ✓ | ✓ |
| Visual Information | ✓ | ✓ | ✗ | ✗ |
| *Task Type* | | | | |
| Question-Answering | ✓ | ✓ | ✗ | ✗ |
| Scientific Computing | ✓ | ✗ | ✓ | ✓ |
| GUI Automation | ✓ | ✗ | ✗ | ✗ |

Table 7: A comparison of SCIENCEBOARD to notable and recent AI4Science benchmarks.

agents to fulfill tasks like humans do. Additionally, SCIENCEBOARD covers a broader range of task types compared to existing works, including but not limited to question-answering and scientific computing. These unique features make SCIENCEBOARD both a versatile playground and an expandable framework for evaluating agents' scientific capabilities.

### D.4 MORE EVALUATION SCRIPT EXAMPLES

Beyond the evaluation cases listed in Section 3.2, Table 8 showcases a broader variety of evaluation pipelines created using our templates.

### D.5 HUMAN PERFORMANCE

In our main experiments, as reflected in Table 3, we recruit college-level students to establish normal human performance on SCIENCEBOARD benchmark. Before attempting the tasks, participants are required to familiarize themselves with foundational knowledge of the relevant scientific disciplines and study the provided operational manuals. They were then given instructions, as shown in Instruction 1, to complete the assigned tasks. Participants were compensated at a rate of $10 per hour for their involvement.

The SCIENCEBOARD environment and scientific software used do not record any personal information, and all participants provide informed consent. The experiment does not involve surveys, interviews, or any behavioral tracking.

### D.6 STABILITY ANALYSIS

Considering that dynamic environments could potentially lead to experimental instability, we conduct an additional set of experiments focusing on consistency. For these, we utilize `GPT-4o` under the `allytree` + screenshot setting, with results and error bars reported in Figure 7.

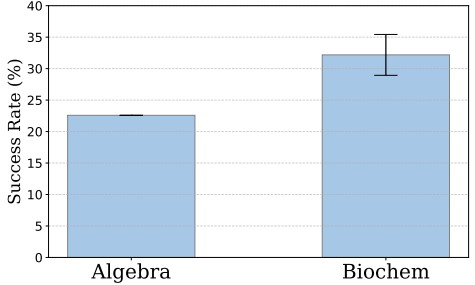

Figure 7: Stability analysis.

Across three independent runs, performance on Algebra tasks remains stable. However, Biochemistry tasks exhibited minor fluctuations in success rates. Upon closer inspection of individual cases, we hypothesize that these variations likely stem from network connectivity issues or transient system lag encountered during task execution.

Table 8: More evaluation cases of SCIENCEBOARD include exact matching, range-based assessment, and numerical tasks with tolerance.

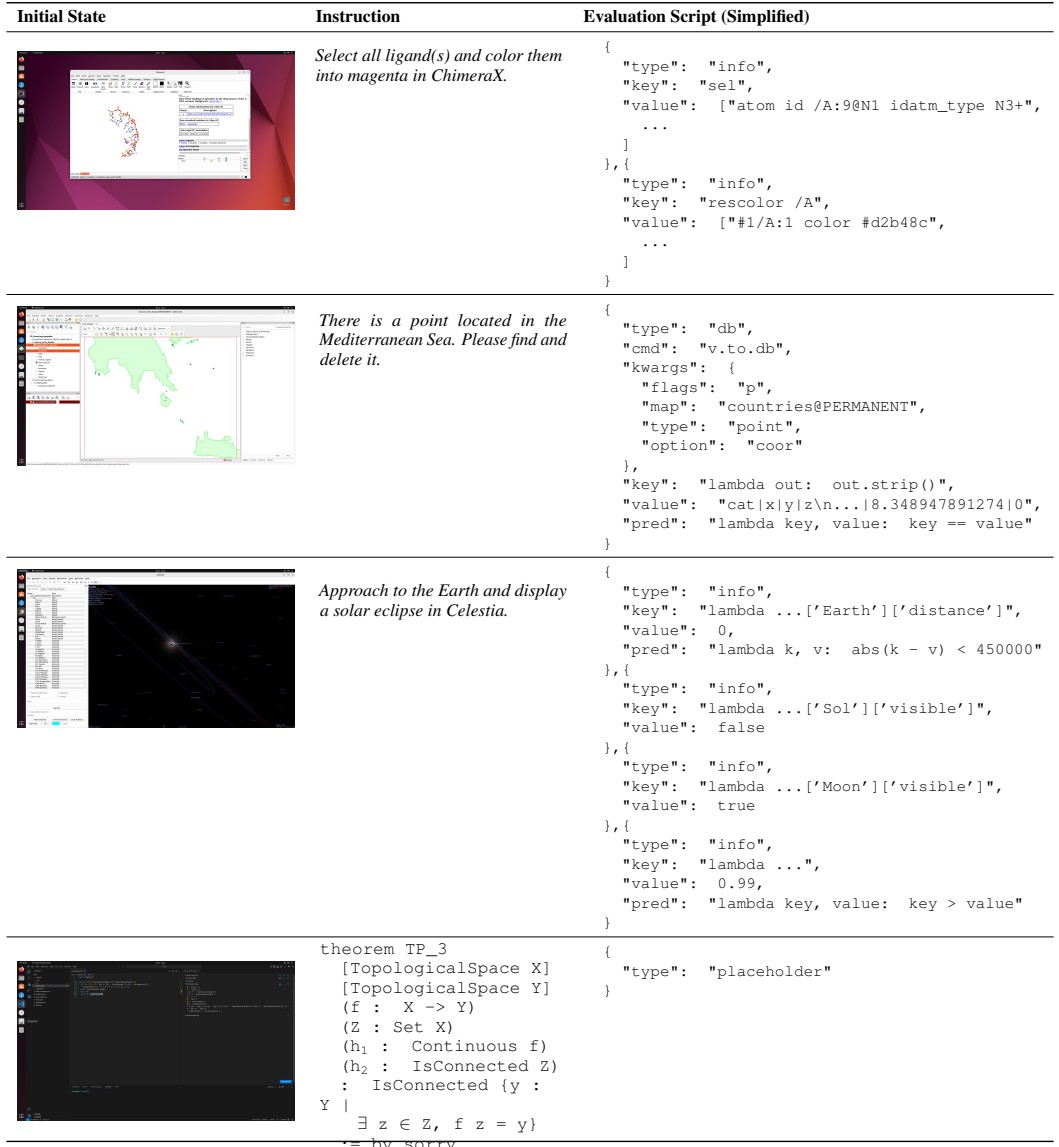

| Initial State | Instruction | Evaluation Script (Simplified) |
|---|---|---|
| | *Select all ligand(s) and color them into magenta in ChimeraX.* | ```{ "type": "info", "key": "sel", "value": ["atom id /A:9@N1 idatm_type N3+", ... ] },{ "type": "info", "key": "rescolor /A", "value": ["#1/A:1 color #d2b48c", ... ] }``` |
| | *There is a point located in the Mediterranean Sea. Please find and delete it.* | ```{ "type": "db", "cmd": "v.to.db", "kwargs": { "flags": "p", "map": "countries@PERMANENT", "type": "point", "option": "coor" }, "key": "lambda out: out.strip()", "value": "cat|x|y|z\n...|8.348947891274|0", "pred": "lambda key, value: key == value" }``` |
| | *Approach to the Earth and display a solar eclipse in Celestia.* | ```{ "type": "info", "key": "lambda ...['Earth']['distance']", "value": 0, "pred": "lambda k, v: abs(k - v) < 450000" },{ "type": "info", "key": "lambda ...['Sol']['visible']", "value": false },{ "type": "info", "key": "lambda ...['Moon']['visible']", "value": true },{ "type": "info", "key": "lambda ...", "value": 0.99, "pred": "lambda key, value: key > value" }``` |
| | ```theorem TP_3 [TopologicalSpace X] [TopologicalSpace Y] (f : X -> Y) (Z : Set X) (h₁ : Continuous f) (h₂ : IsConnected Z) : IsConnected {y : Y | ∃ z ∈ Z, f z = y} := by sorry``` | ```{ "type": "placeholder" }``` |

## D.7 EVALUATION COST

We use API keys to access proprietary models. On average, a single run on all SCIENCE-BOARD tasks costs $64 using `GPT-4o`, $86 using `Claude-3.7-Sonnet`, and $45 using `Gemini-2.0-Flash`.

## E DETAILS OF EXPERIMENTS

### E.1 BACKBONE MODELS

We briefly discuss the backbones we used to build our computer-using agents.

**Proprietary Models.** Proprietary models now demonstrate striking capabilities in complex reasoning and are increasingly exhibiting agentic potential for dynamic real-world interaction, prompting a

closer look at their diverse forms. In the experimental section, we accessed the following proprietary models via API keys:

- `GPT-4o` (Hurst et al., 2024).
- `GPT-5` (OpenAI, 2025b).
- `Claude-3.7-Sonnet` (Anthropic AI, 2024).
- `Gemini-2.0-Flash` (Gemini Team, 2024).
- `Gemini-2.5-Pro` (Gemini Team, 2025)
- `o3-mini` (OpenAI, 2025a).

**Open-source Models.** Open-source models are demonstrating remarkable advancements, steadily narrowing the performance gap with proprietary models. Crucially, the open-source community recognized the significance of agentic capabilities early on, fostering development in this direction. This foresight has translated into exceptional performance, particularly within GUI scenarios where these models now excel on various challenging benchmarks. Our evaluation is based on the following open-source models, which are characterized by their advanced grounding capabilities:

- Qwen2.5-VL-72B-Instruct (Bai et al., 2025): The latest evolution in the Qwen vision-language model family, primarily distinguished by its robust agentic capabilities. It operates directly as a visual agent, proficient in reasoning, dynamically utilizing tools, and executing tasks for computer and phone operation. Complementing its agentic prowess, Qwen2.5-VL-72B-Instruct demonstrates advanced proficiency in detailed visual analysis (including texts, charts, icons, and layouts within images), comprehension of videos exceeding one hour with event pinpointing, precise object localization with structured coordinate output, and the generation of structured data from documents such as invoices and forms. In our experiments, this model is deployed using interconnected clusters of $8 \times$ A100 80GB GPUs with vLLM (Kwon et al., 2023).

- InternVL3-78B (Chen et al., 2024): An advanced MLLM recognized for its superior overall performance and significantly enhanced multimodal perception and reasoning. A key advancement is its robust agentic functionality, demonstrated through proficient tool usage and GUI agent operations, alongside extended capabilities in areas like industrial image analysis and 3D vision perception. These comprehensive abilities are underpinned by innovations such as a native multimodal pre-training approach, supervised fine-tuning with diverse, high-quality data tailored to these advanced tasks, and mixed preference optimization for refined reasoning. In our experiments, this model is deployed using interconnected clusters of $8 \times$ A100 80GB GPUs with vLLM.

- QvQ-72B-Preview (Qwen Team, 2024): An experimental research model focused on advancing visual reasoning capabilities. It has achieved compelling performance in complex multidisciplinary understanding and problem-solving, highlighting its specialized strength in sophisticated visual cognitive tasks. However, it exhibits some limitations in instruction following, appearing less adept in agent scenarios that require precise action outputs. In our experiments, this model is deployed using interconnected clusters of $8 \times$ A100 80GB GPUs with vLLM.

**GUI Action Models.** While foundational models provide impressive general-purpose intelligence, their intrinsic agentic capabilities for nuanced GUI manipulation are still under active exploration, often requiring further specialization. Consequently, a prominent line of research involves adapting open-source VLMs by fine-tuning them on extensive, GUI-specific datasets. This targeted training methodology yields dedicated action models equipped with significantly enhanced proficiencies for understanding and interacting with GUIs. The GUI action models adopted in this paper are as follows:

- OS-Atlas-Pro-7B (Wu et al., 2025b): A foundational GUI action model that significantly advances open-source VLMs for agentic tasks, excelling in GUI grounding and out-of-distribution scenarios through innovations in modeling and the creation of the largest open-source, cross-platform GUI grounding corpus with over 13 million elements. It demonstrates state-of-the-art performance across six diverse benchmarks (mobile, desktop, web) and verifies the existence of model scaling laws in GUI scenarios. In our experiments, this model is deployed using a single A100 80GB GPU with vLLM (Kwon et al., 2023).

- UGround-V1-7B (Gou et al., 2025): A universal visual grounding model that identifies GUI action elements by pixel coordinates. It powers the SeeAct-V framework (Zheng et al., 2024), which enables purely visual GUI perception and pixel-level operations. Agents using SeeAct-V with UGround have achieved SOTA results across five distinct benchmarks spanning web, mobile, and desktop evaluations. In our experiments, this model is deployed on a single A100 80GB GPU with vLLM.

- UI-TARS-72B-DPO (Qin et al., 2025): An end-to-end native GUI agent that uniquely perceives screenshots as its sole input to perform human-like keyboard and mouse interactions, outperforming prevailing agent frameworks that depend on heavily wrapped commercial models with expert-crafted prompts. It has established state-of-the-art performance across more than ten GUI agent benchmarks. This advanced capability stems from key innovations including enhanced perception, unified action modeling, System-2 reasoning, iterative training with reflective online traces, and a final Direct Preference Optimization (DPO) phase, which refines its ability to make precise, context-aware decisions. In our experiments, UI-TARS-72B-DPO utilizes vLLM for inference and is deployed on interconnected clusters of $8 \times$ A100 80GB GPUs.

- GUI-Actor-7B (Wu et al., 2025a): A recently proposed GUI grounding model that introduces a novel coordinate-free visual grounding approach. It utilizes an action head to direct the special token <ACTOR> to the target screenshot patches for localization. It claims to surpass the text-based coordinate prediction baseline and demonstrates better generalization in out-of-distribution (OOD) scenarios. In our experiments, we used the 7B version of GUI-Actor based on the Qwen2.5-VL backbone.

## E.2 Evaluation Settings - Main Experiments

We adhered to common prompt engineering strategies from previous works (Sun et al., 2024b; Zhou et al., 2024; Zhang & Zhang, 2024) for the agents under evaluation. For each domain, the agent interacts with the environment under the guidance of a meta-prompt, which includes information about the software being operated, executable special actions, and related details. When taking actions, the agent generates outputs in the ReAct style (Yao et al., 2023), with its step-by-step thoughts recorded in the interaction history.

Throughout the evaluation, we set the `temperature parameter` to 0.5, `top_p` to 0.9, and `max_tokens` to 1500. We list some prompt examples in Prompt 14, Prompt 15, Prompt 16 and Prompt 17.

## E.3 Evaluation Settings - Analysis

In experiments with interleaved planning and action, we first address inconsistencies in coordinate outputs from different GUI action models. While InternVL3-78B (Chen et al., 2024) outputs coordinates on a `[0, 1]` scale, models such as OS-Atlas, UI-TARS, and UGround use a `[0, 1000]` scale. To ensure uniformity, we normalized all coordinate outputs to a `[0, 1]` scale prior to execution.

This part of the experiments employs a two-stage process: First, the planner model receives the current observation (obs) and task instruction to generate a high-level plan or a specific action. If the planner outputted a directly executable primitive action (*e.g.*, a non-GUI system-level command or a special control token like `DONE`), that action will be performed immediately, and the action model was not invoked for that step. Otherwise, the grounding model received the current observation and the plan (or sub-task) from the planner. Its role was to output low-level executable instructions. If the grounding model generate `pyautogui` actions directly, these commands were executed. For models outputting in their specific native formats, we implement custom parsers to translate these into `pyautogui` actions: for UGround and UI-TARS, all coordinate-based outputs were interpreted as `click`, whereas for OS-Atlas, its outputs were parsed to differentiate between `click`, `type`, and `scroll` based on its defined schema.

We list some prompt examples in Prompt 18, Prompt 19, Prompt 20 and Prompt 21.

# F EXTENDED ANALYSIS

## F.1 INTERFACES

In Section 6, we analyze the performance difference between Vision-Only and Hybrid Interface settings under the `a11ytree` + screenshot. Here, we present empirical results under the other three observation settings.

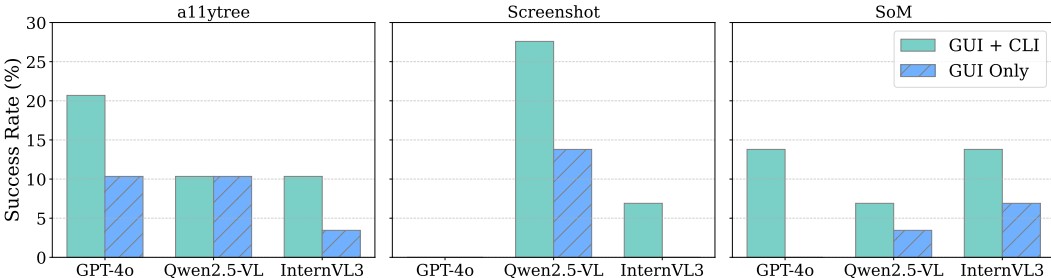

Figure 8: Extended analysis of Vision-Only vs. Hybrid Interface.

As shown in Figure 8, the hybrid GUI + CLI setting consistently achieves performance that is comparable to or better than the GUI-Only setting across all scenarios. Interestingly, while `GPT-4o` achieves state-of-the-art performance under other observation settings, it exhibits very weak action capabilities when using screenshot setting, indicating the reliance on structured observations for effective reasoning and planning.

## F.2 INTERACTIVE ENVIRONMENTS

ATP represents one of the most logic-intensive tasks for agents and has been traditionally studied in textual settings in prior works (*e.g.*, plain text or bash terminal).

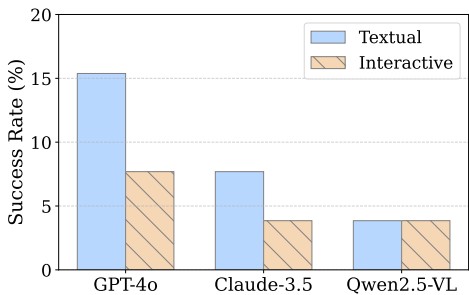

Figure 9: Textual v.s. Interactive

We extend ATP to live OS in SCIENCEBOARD and further compare agents' performance under textual and interactive settings. The latter, similar to environments commonly used by humans, provides a live VSCode interface with features such as syntax highlighting, autocompletion, type inference, and other functionalities. As shown in Figure 9, in the textual setting, the agent applies heuristic strategies (*e.g.*, Monte Carlo search) to make predictions over the proof tree without interacting with the environment. In contrast, in the interactive setting, the agent must autonomously decide which PROOFSTATE to proceed with. Moreover, the agent is also required to localize the relevant code segments within the interface. Completing formal methods tasks becomes substantially more challenging in realistic environments, which significantly increases the cognitive complexity.

## F.3 DIFFICULTY ANALYSIS

We further analyze the success rates of computer-using agents on the SCIENCEBOARD benchmark across different task difficulty levels. We employ `Claude-3.7-Sonnet`, `GPT-4o`, and Qwen2.5-VL, with results presented in Figure 10.

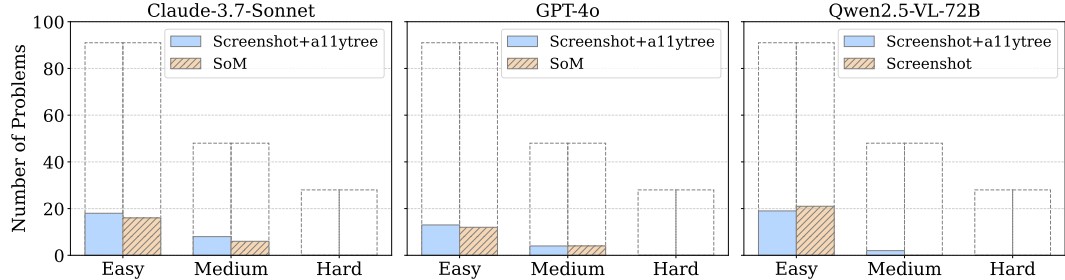

Figure 10: Comparative analysis of task difficulty solve rates.

The findings indicate that solvable tasks are primarily concentrated among a subset of "Easy" problems and a few "Medium" tasks. All "hard" tasks, which involve complex computations, cross-application workflows, or long-horizon planning, could not be completed by any of the evaluated agents.

## F.4 COMPUTE SCALING

During the evaluation of frontier models, a natural question arises: can scaling inference-time compute (test-time scaling) resolve the performance bottlenecks observed in complex scientific workflows? To investigate whether extended generation limits and native reasoning modes enable agents to better navigate these dynamic environments, we conducted an extended analysis focusing on three domains.

Table 9: Success rates of varying `max_tokens` under the Screenshot observation setting.

| max_tokens | Algebra | Biochemistry | GIS |
|---|---|---|---|
| 1500 | 41.90% | 62.10% | 11.80% |
| 2000 | 41.90% | 58.62% | 11.80% |
| 2500 | 41.90% | 65.52% | 14.70% |

Further, we evaluated the performance of `GPT-5` utilizing its native reasoning mode with varying levels of test-time reasoning effort (`medium` and `high`). As detailed in Table 10, elevating the reasoning effort improves performance across several settings. Notably, under the *Screenshot + a11ytree* setting, shifting from medium to high reasoning effort slightly increases the success rate in Biochemistry from 68.96% to 72.41%, and in GIS from 14.70% to 17.64%.

Table 10: Performance of `GPT-5` under different native reasoning efforts across two observation settings.

| Observation | Reasoning Effort | Algebra | Biochemistry | GIS |
|---|---|---|---|---|
| Screenshot | Medium | 41.90% | 65.52% | 14.70% |
| | High | 45.16% | 65.52% | 14.70% |
| Screenshot + a11ytree | Medium | 48.39% | 68.96% | 14.70% |
| | High | 51.61% | 72.41% | 17.64% |

These results empirically demonstrate that while scaling test-time compute and leveraging native reasoning models provide measurable benefits, they do not fundamentally overcome the core challenges of our benchmark. The bottlenecks likely stem from the agent's general agentic capabilities, *i.e.*, the ability to accurately perceive dense, domain-specific UI elements and translate high-level scientific plans into precise, executable actions, rather than solely a deficiency in deep cognitive reasoning.

## F.5 FAILURE ANALYSIS

To further investigate the reasons why computer-using agents fail when planning or taking actions on scientific tasks, here we include and discuss several typical examples of such errors.

**Opening the Wrong File.** This error is frequently caused by grounding issues. The agent initially clicks on an incorrect file and then attempts to perform subsequent actions, such as inputting data, within that wrong file. This often leads to the agent repeatedly making the same mistake or getting stuck in an unproductive loop. A typical case is shown in Figure 11.

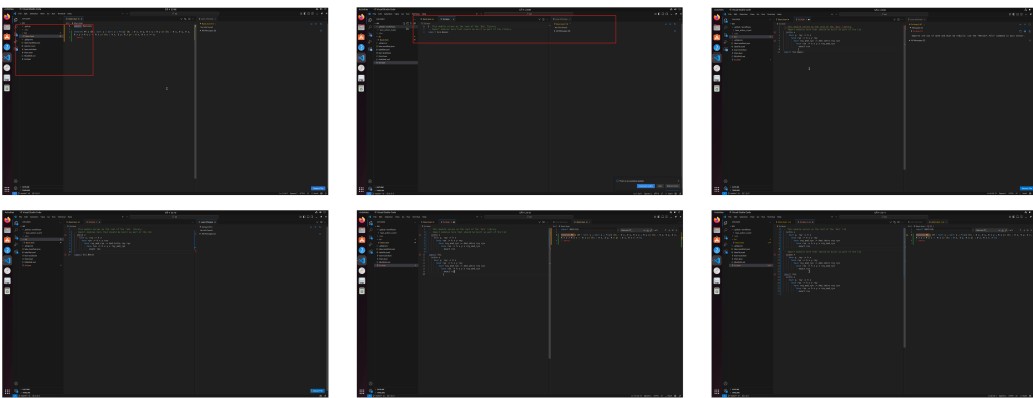

Figure 11: Use wrong file.

**Inability to Invoke the Correct Function.** In some instances, agents need to identify and use a specific function within a software application but attempt to do so by directly typing an assumed function name into a search bar or command input. If the exact function name is unknown or guessed incorrectly, a more robust strategy would be to browse available menus or function lists. Instead, agents may incorrectly assume knowledge of the function name and attempt to look up its usage, leading to failure. A typical example of this behavior is presented in Figure 12.

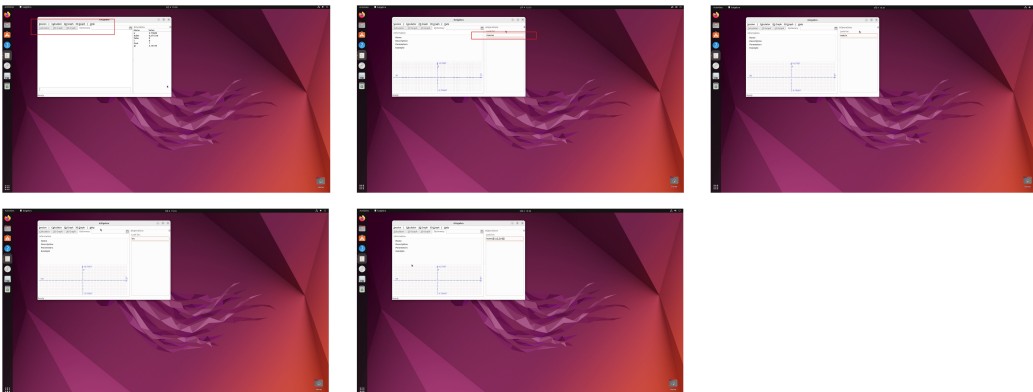

Figure 12: Function invocation error.

**Incorrect CLI Code.** Failures also occur when agents formulate CLI commands incorrectly. This can involve syntax errors, wrong command names, or incorrect parameters. Notably, in some of these failed CLI attempts, the intended task could have been accomplished more straightforwardly by interacting with a corresponding button or element in the GUI. A typical example is shown in Figure 13.

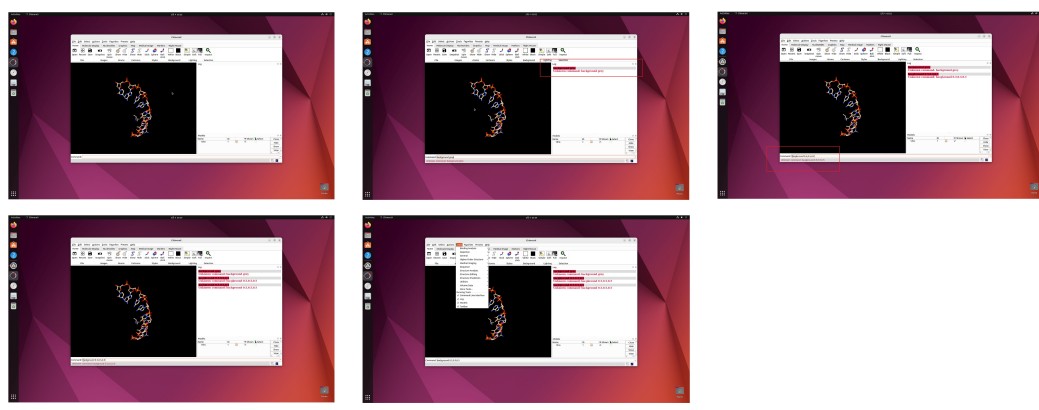

Figure 13: CLI code error.

# G    PROMPTS

The prompt examples we used in SCIENCEBOARD are listed below.

```
Agentic Prompt - ChimeraX with screenshot

You are an agent which follow my instruction and perform desktop computer
tasks as instructed.
You have good knowledge of ChimeraX, a molecular visualization software;
and assume your code will run on a computer controlling the mouse and
keyboard.
For each step, you will get an observation of the desktop by an
accessibility tree, which is based on AT-SPI library, and you will
predict actions of the next step based on that.

You are required to use `pyautogui` to perform the action grounded to the
observation, but DO NOT use the `pyautogui.locateCenterOnScreen` function
to locate the element you want to operate with since we have no image of
the element you want to operate with.  DO NOT USE `pyautogui.screenshot()`
to make screenshot.
You ONLY need to return the code inside a code block, like this:
``
# your code here
``
Return one line or multiple lines of python code to perform the action
each time, and be time efficient.  When predicting multiple lines of
code, make some small sleep like `time.sleep(0.5);` interval so that the
machine could take breaks.  Each time you need to predict a complete code,
and no variables or function can be shared from history.

Specially, it is also allowed to return the following special code:
When you think the task is done, return "`DONE`";
When you think the task can not be done, return "`FAIL`".  Don't easily
say "`FAIL`"; try your best to do the task;
When you think you have to wait for some time, return "`WAIT`" or "`WAIT
n`", in which n defaults to 5(s);
When you are asked to submit an answer, return "`ANS s`" without
quotation marks surrounding s, and use `FAIL` if there is no answer to
the question.

My computer's password is 'password', feel free to use it when you need
sudo rights.
DO NOT introduce any unrelated models or easily close existing models,
otherwise the task might be evaluated as FAILED.
DO NOT close the current ChimeraX session, or every effort you made will
be in vain.
NEVER try to reopen the command line interface in ChimeraX if it is
hidden, because it has been deactivated and cannot do anything.  But you
are welcome to use it once it is presented.

First give the current observation and previous things we did a short
reflection, then RETURN ME THE CODE OR SPECIAL CODE I ASKED FOR. NEVER
EVER RETURN ME ANYTHING ELSE.
You are asked to complete the following task:  Fetch 2OLX from PDB in
ChimeraX.
```

Prompt 14: Prompts for ChimeraX with screenshot

**Agentic Prompt – Celestia with screenshot**

You are an agent which follow my instruction and perform desktop computer tasks as instructed.
You have good knowledge of Celestia, a three-dimension space simulator; and assume your code will run on a computer controlling the mouse and keyboard.
For each step, you will get an observation of the desktop by a screenshot, and you will predict actions of the next step based on that.

You are required to use `pyautogui` to perform the action grounded to the observation, but DO NOT use the `pyautogui.locateCenterOnScreen` function to locate the element you want to operate with since we have no image of the element you want to operate with. DO NOT USE `pyautogui.screenshot()` to make screenshot.
You ONLY need to return the code inside a code block, like this:
```
# your code here
```
Return one line or multiple lines of python code to perform the action each time, and be time efficient. When predicting multiple lines of code, make some small sleep like `time.sleep(0.5);` interval so that the machine could take breaks. Each time you need to predict a complete code, and no variables or function can be shared from history.

Specially, it is also allowed to return the following special code:
When you think the task is done, return "`DONE`";
When you think the task can not be done, return "`FAIL`". Don't easily say "`FAIL`"; try your best to do the task;
When you think you have to wait for some time, return "`WAIT`" or "`WAIT n`", in which n defaults to 5(s);
When you are asked to submit an answer, return "`ANS s`" without quotation marks surrounding s, and use `FAIL` if there is no answer to the question.

My computer's password is 'password', feel free to use it when you need sudo rights.
The criterion for a celestial body to be displayed on the screen is that the object's center is within the window range and is not blocked by others.

First give the current observation and previous things we did a short reflection, then RETURN ME THE CODE OR SPECIAL CODE I ASKED FOR. NEVER EVER RETURN ME ANYTHING ELSE.
You are asked to complete the following task: Set the Julian date to 2400000 in Celestia.

Prompt 15: Prompts for Celestia with screenshot

```
Agentic Prompt - ChimeraX with set-of-marks

You are an agent which follow my instruction and perform desktop computer
tasks as instructed.
You have good knowledge of ChimeraX, a molecular visualization software;
and assume your code will run on a computer controlling the mouse and
keyboard.
For each step, you will get an observation of the desktop by 1) an
accessibility tree, which is based on AT-SPI library; and 2) a screenshot
with interact-able elements marked with numerical tags, and you will
predict actions of the next step based on that.

You are required to use `pyautogui` to perform the action grounded to the
observation, but DO NOT use the `pyautogui.locateCenterOnScreen` function
to locate the element you want to operate with since we have no image of
the element you want to operate with.  DO NOT USE `pyautogui.screenshot()`
to make screenshot.
You ONLY need to return the code inside a code block, like this:
```
# your code here
```
Return one line or multiple lines of python code to perform the action
each time, and be time efficient.  When predicting multiple lines of
code, make some small sleep like `time.sleep(0.5);` interval so that the
machine could take breaks.  Each time you need to predict a complete code,
and no variables or function can be shared from history.

You can replace x, y in the code with the tag of elements you want to
operate with, such as:
```
pyautogui.moveTo(tag_3)
pyautogui.click(tag_2)
pyautogui.dragTo(tag_1, button='left')
```
When you think you can directly output precise x and y coordinates or
there is no tag on which you want to interact, you can also use them
directly; but you should be careful to ensure the correct of coordinates.

Specially, it is also allowed to return the following special code:
When you think the task is done, return "`DONE`";
When you think the task can not be done, return "`FAIL`".  Don't easily
say "`FAIL`"; try your best to do the task;
When you think you have to wait for some time, return "`WAIT`" or "`WAIT
n`", in which n defaults to 5(s);
When you are asked to submit an answer, return "`ANS s`" without
quotation marks surrounding s, and use `FAIL` if there is no answer to
the question.

My computer's password is 'password', feel free to use it when you need
sudo rights.
DO NOT introduce any unrelated models or easily close existing models,
otherwise the task might be evaluated as FAILED.
DO NOT close the current ChimeraX session, or every effort you made will
be in vain.
NEVER try to reopen the command line interface in ChimeraX if it is
hidden, because it has been deactivated and cannot do anything.  But you
are welcome to use it once it is presented.

First give the current observation and previous things we did a short
reflection, then RETURN ME THE CODE OR SPECIAL CODE I ASKED FOR. NEVER
EVER RETURN ME ANYTHING ELSE.
You are asked to complete the following task:  Fetch 2OLX from PDB in
ChimeraX.
```

Prompt 16: Prompts for ChimeraX with Set-of-Marks

```
Agentic Prompt - Celestia with set-of-marks

You are an agent which follow my instruction and perform desktop computer
tasks as instructed.
You have good knowledge of Celestia, a three-dimension space simulator;
and assume your code will run on a computer controlling the mouse and
keyboard.
For each step, you will get an observation of the desktop by 1) an
accessibility tree, which is based on AT-SPI library; and 2) a screenshot
with interact-able elements marked with numerical tags, and you will
predict actions of the next step based on that.

You are required to use 'pyautogui' to perform the action grounded to the
observation, but DO NOT use the 'pyautogui.locateCenterOnScreen' function
to locate the element you want to operate with since we have no image of
the element you want to operate with.  DO NOT USE 'pyautogui.screenshot()'
to make screenshot.
You ONLY need to return the code inside a code block, like this:
```
# your code here
```
Return one line or multiple lines of python code to perform the action
each time, and be time efficient.  When predicting multiple lines of
code, make some small sleep like 'time.sleep(0.5);' interval so that the
machine could take breaks.  Each time you need to predict a complete code,
and no variables or function can be shared from history.

You can replace x, y in the code with the tag of elements you want to
operate with, such as:
```
pyautogui.moveTo(tag_3)
pyautogui.click(tag_2)
pyautogui.dragTo(tag_1, button='left')
```
When you think you can directly output precise x and y coordinates or
there is no tag on which you want to interact, you can also use them
directly; but you should be careful to ensure the correct of coordinates.

Specially, it is also allowed to return the following special code:
When you think the task is done, return "'DONE'";
When you think the task can not be done, return "'FAIL'".  Don't easily
say "'FAIL'"; try your best to do the task;
When you think you have to wait for some time, return "'WAIT'" or "'WAIT
n'", in which n defaults to 5(s);
When you are asked to submit an answer, return "'ANS s'" without
quotation marks surrounding s, and use 'FAIL' if there is no answer to
the question.

My computer's password is 'password', feel free to use it when you need
sudo rights.
The criterion for a celestial body to be displayed on the screen is that
the object's center is within the window range and is not blocked by
others.

First give the current observation and previous things we did a short
reflection, then RETURN ME THE CODE OR SPECIAL CODE I ASKED FOR. NEVER
EVER RETURN ME ANYTHING ELSE.
You are asked to complete the following task:  Set the Julian date to
2400000 in Celestia.
```

Prompt 17: Prompts for Celestia with Set-of-Marks

```
Human Instructions

You are required to finish the given tasks manually to provide sample
data of human accuracy.
First, please start up the evaluation script with debug option ON and
headless option OFF. Then, wait for the environment to be initialized
and perform your actions when you receive corresponding logs from stdout.
Press ENTER after you finish operating and the script will evaluate your
result submitted automatically.
Attention:
1.  If you need to finish the task with primitives other than TIMEOUT,
please input directly into stdin;
2.  You can search for documents or manuals if you encounter
domain-specific knowledge you are not familiar with;
3.  Make sure that the number of your steps is less than expected.  To be
more precise, a popup without possibility to predict its position should
be split into different steps.
```

Instruction 1: Instruction for humans.

```
Agentic Prompt - OS-Atlas

You are an agent which follow my instruction and perform desktop computer
tasks as instructed.
You have good knowledge of Celestia, a three-dimension space simulator;
and assume your code will run on a computer controlling the mouse and
keyboard.
For each step, you will get an observation of the desktop by a screenshot,
together with a plan generated by the planner, and you will parse the
plan to operate actions of next steps based on that.

You are required to use your grounding ability to perform the action
grounded to the observation and the plan.
You need to return a basic action together with arguments, of which the
available ones are listed below:
CLICK: to click at the specified position.
   - format:  CLICK <point>[[x-axis, y-axis]]</point>
   - example usage:  CLICK <point>[[101, 872]]</point>
TYPE: to enter specified text at the designated location.
   - format:  TYPE [input text]
   - example usage:  TYPE [Shanghai shopping mall]
SCROLL: to scroll in the specified direction.
   - format:  SCROLL [direction (UP/DOWN/LEFT/RIGHT)]
   - example usage:  SCROLL [UP]

My computer's password is 'password', feel free to use it when you need
sudo rights.
Some plans provided may contains unexpected code blocks or confusing
instructions.  Be flexible and adaptable according to changing
circumstances.

First give the current observation and the generated plan, then RETURN
ME THE CODE I ASKED FOR. NEVER EVER RETURN ME ANYTHING ELSE.
You are asked to complete the following task:  Set the Julian date to
2400000 in Celestia.
```

Prompt 18: Prompts for OS-Atlas

```
Agentic Prompt - UGround

You are an agent which follow my instruction and perform desktop computer
tasks as instructed.
You have good knowledge of Celestia, a three-dimension space simulator;
and assume your code will run on a computer controlling the mouse and
keyboard.
For each step, you will get an observation of the desktop by a screenshot,
together with a plan generated by the planner, and you will parse the
plan to operate actions of next steps based on that.

You are required to use your grounding ability to perform the action
grounded to the observation and the plan.
You need to return a 2d coordinate (x, y) indicating the position you
want to click.

My computer's password is 'password', feel free to use it when you need
sudo rights.
Some plans provided may contains unexpected code blocks or confusing
instructions.  Be flexible and adaptable according to changing
circumstances.

First give the current observation and the generated plan, then RETURN
ME THE CODE I ASKED FOR. NEVER EVER RETURN ME ANYTHING ELSE.
You are asked to complete the following task:  Set the Julian date to
2400000 in Celestia.
```

Prompt 19: Prompts for UGround

```
Agentic Prompt – Qwen

You are an agent which follow my instruction and perform desktop computer
tasks as instructed.
You have good knowledge of Celestia, a three-dimension space simulator;
and assume your code will run on a computer controlling the mouse and
keyboard.
For each step, you will get an observation of the desktop by a screenshot,
together with a plan generated by the planner, and you will parse the
plan to operate actions of next steps based on that.

You are required to use `pyautogui` to perform the action
grounded to the observation and the plan, but DO NOT use the
`pyautogui.locateCenterOnScreen` function to locate the element you want
to operate with since we have no image of the element you want to operate
with.  DO NOT USE `pyautogui.screenshot()` to make screenshot.
You ONLY need to return the code inside a code block, like this:
```
# your code here
```
Return one line or multiple lines of python code to perform the action
each time, and be time efficient.  When predicting multiple lines of
code, make some small sleep like `time.sleep(0.5);` interval so that the
machine could take breaks.  Each time you need to predict a complete code,
and no variables or function can be shared from history.

Specially, it is also allowed to return the following special code:
When you think the task is done, return "`DONE`";
When you think the task can not be done, return "`FAIL`".  Don't easily
say "`FAIL`"; try your best to do the task;
When you think you have to wait for some time, return "`WAIT`" or "`WAIT
n`", in which n defaults to 5(s);
When you are asked to submit an answer, return "`ANS s`" without
quotation marks surrounding s, and use `FAIL` if there is no answer to
the question.

My computer's password is 'password', feel free to use it when you need
sudo rights.
Some plans provided may contains unexpected code blocks or confusing
instructions.  Be flexible and adaptable according to changing
circumstances.

First give the current observation and the generated plan, then RETURN
ME THE CODE OR SPECIAL CODE I ASKED FOR. NEVER EVER RETURN ME ANYTHING
ELSE.
You are asked to complete the following task:  Set the Julian date to
2400000 in Celestia.
```

Prompt 20: Prompts for Qwen

```
Agentic Prompt - UI-Tars

You are an agent which follow my instructions and performs desktop
computer tasks as instructed.
You have good knowledge of Celestia, a three-dimension space simulator;
and assume your code will run on a computer controlling the mouse and
keyboard.
For each step, you will get an observation of the desktop by a screenshot,
together with a plan generated by the planner, and you will parse the
plan to operate actions of next steps based on that.

You are required to use your grounding ability to perform the action
grounded to the observation and the plan.
You need to return a 2d coordinate (x, y) indicating the position you
want to click.

My computer's password is 'password', feel free to use it when you need
sudo rights.
Some plans provided may contains unexpected code blocks or confusing
instructions.  Be flexible and adaptable according to changing
circumstances.

First give the current observation and the generated plan, then RETURN
ME THE CODE I ASKED FOR. NEVER EVER RETURN ME ANYTHING ELSE.
You are asked to complete the following task:  Set the Julian date to
2400000 in Celestia.
```

Prompt 21: Prompts for UI-TARS

