# OpenReview forum: "ScienceBoard: Evaluating Multimodal Autonomous Agents in Realistic Scientific Workflows"
_ICLR.cc/2026/Conference — ICLR 2026 Poster_

### Official Review · Reviewer_N2NY · 2025-10-31

**Soundness:** 4
**Presentation:** 4
**Contribution:** 4
**Rating:** 6
**Confidence:** 5

**Summary:**

This paper introduces SciBoard, a benchmark designed to evaluate LLM/VLM-based agents on scientific-discovery workflows in the following domains: biochemistry, algebra, theorem proving, geoinformatics, astronomy, and scientific documentation. The authors created six environments as virtual machines that can be interacted with via command lines, APIs, or the GUI. Across all environments, SciBoard regroup set of 169 tasks designed by domain experts. Each task provides specific instructions and a mean to automatically evaluate its completion. The authors empirically show how challenging SciBoard is for current LLM/VLM-based agents achieving only a 15% overall success rate.

**Strengths:**

- A lot of work went into designing and implementing the six environments for SciBoard. Then, designing the tasks within these environments required careful consideration of the unique challenges and diversity present in each scientific domain.
- The authors show the impact of different type of observations (e.g., images, text, images+text). This provides useful insights into how different modalities contribute to agent capabilities in those particular scientific workflows.

**Weaknesses:**

- I don't see a clear mention of train/validation/test splits for the tasks in SciBoard. Are all 169 tasks used for evaluation only? If so, how do you envision researchers using SciBoard for training their agents? You can bet people will want to use SciBoard for RL fine-tuning their agents on it as well.
- For figure 5, it is mentioned that "GPT-4o and InternVL3 suffer clear drops in performance, whereas Qwen2.5-VL remains largely unaffected, indicating better adaptation to GUI execution". To me, it could also indicate that Qwen2.5-VL is not really leveraging the CLI and only relies on GUI. It would be interesting to see the CLI-only performance for that model as well to confirm that hypothesis.

#### Minor
- How was the human baseline obtained? Was it through domain experts using the same interface as the agents?
- line 376: Gemini-2.0-Flash (Team, 2024) -> (Google, 2024)

**Questions:**

- Is there anything that prevents the agent from cheating given that its has access to the full machine? Could it for instance read some log files or inspect memory to get extra information that a human user wouldn't have access to?
- In table 2, what does the two open problems correspond to exactly? How do you evaluate for this?
- In table 2, in the execution section, what does steps and time correspond to exactly? Are those obtained from human trajectories? Are those human expert or non-expert?

---

> ### Author Response · Authors · 2025-11-19
> **Response to Reviewer N2NY (1/2)**
>
> Thank you for your review. We appreciate your recognizing our significant infrastructure and experimental workload, and highlighting the unique challenges and diversity ScienceBoard benchmark provides across scientific domains.
>
> W1: Are all 169 tasks used for evaluation only? If so, how do you envision researchers using SciBoard for training their agents? You can bet people will want to use SciBoard for RL fine-tuning their agents on it as well.
> Yes, you are correct. Currently, all tasks are reserved for evaluation. This aligns with the practice of other well-established dynamic computer-using environments like OSWorld[1], AndroidWorld[2], and WebArena[3], which also serve primarily as "held-out" evaluation benchmarks.
>
> We appreciate your recognizing the value of our infrastructure for (RL) training, and you are exactly right: **this is a use case we anticipated** (Thus, we designed the infrastructure with this in mind, ensuring the software interfaces and interaction protocols are flexible and robust). *In fact, some works[4] are already utilizing ScienceBoard infrastructure (our interfaces, interaction protocols, and VM environments) to collect their own in-domain trajectories for training their agents*.
>
> Finally, we thank you for your suggestion. We do have some spare tasks from our annotations that are not included in the test set. We plan to release these as a validation set in future updates.
>
>
> > W2: For figure 5, it is mentioned that "GPT-4o and InternVL3 suffer clear drops in performance, whereas Qwen2.5-VL remains largely unaffected, indicating better adaptation to GUI execution". To me, it could also indicate that Qwen2.5-VL is not really leveraging the CLI and only relies on GUI. It would be interesting to see the CLI-only performance for that model as well to confirm that hypothesis.
>
> We appreciate this insightful comment. While it is a good hypothesis that stability might imply a lack of CLI usage, our extended analysis in Appendix F.1 (Figure 8) already provides evidence!
>
> Specifically, under the "Screenshot" observation setting, Qwen2.5-VL shows a significant performance drop (from ~28% -> ~14%) when the CLI interface is removed. This demonstrates that Qwen2.5-VL can leverage CLI under settings beyond a11ytree + screenshot.
>
> Regarding the "CLI-only": Across the diverse software covered in this work, GUI and CLI actions do not map equivalently. For most software (except ChimeraX), the CLI only covers a subset of functionalities, whereas the GUI provides complete access. Consequently, a general "CLI-only" evaluation would make many tasks inherently unsolvable.
>
> Due to the page limits of submission, detailed results on these GUI/CLI dynamics were placed in the appendix. If additional space is given, we will follow our discussion to expand this in the main body.
>
>
> > Q1: Is there anything that prevents the agent from cheating given that its has access to the full machine? Could it for instance read some log files or inspect memory to get extra information that a human user wouldn't have access to?
>
> Thank you for this insightful question. Our agent's access generally mirrors a human user, who also interacts with the full OS and file system; this capability is necessary for our tasks.
>
> The key safeguard is: our evaluation method is programmatic and detached from the VM. Success is judged by state-checking functions that verify concrete outcomes (e.g., whether the result itself is within tolerance, rather than relying on other information extracted from the machine). This makes the risk of the agent hacking the test, rather than genuinely solving it, very low.
>
> ### References
>
> [1] OSWorld: Benchmarking Multimodal Agents for Open-Ended Tasks in Real Computer Environments https://arxiv.org/abs/2404.07972
>
> [2] AndroidWorld: A Dynamic Benchmarking Environment for Autonomous Agents https://arxiv.org/abs/2405.14573
>
> [3] WebArena: A Realistic Web Environment for Building Autonomous Agents https://arxiv.org/abs/2307.13854
>
> [4] CODA: Coordinating the Cerebrum and Cerebellum for a Dual-Brain Computer Use Agent with Decoupled Reinforcement Learning https://arxiv.org/abs/2508.20096

---

> > ### Author Response · Authors · 2025-11-19
> > **Response to Reviewer N2NY (2/2)**
> >
> > > Q2: In table 2, what does the two open problems correspond to exactly? How do you evaluate for this?
> >
> > These two "Open Problems" represent our specific initiative to evaluate agents on "discovering new science." They require the agent to leverage its full range of domain knowledge, tools, and programming to tackle two famous unsolved conjectures: the Sum of Four Cubes problem and Chvátal's conjecture.
> >
> > Unlike standard tasks, these open exploration problems lack a known ground truth. To evaluate them objectively, we employ formal verification with to convert the challenge into a deterministic, programmatic one, for example: `def OP_1 (n : ℤ) : Decidable (∃ a b c d : ℤ, n = a ^ 3 + b ^ 3 + c ^ 3 + d ^ 3) := by sorry` (Sum of Four Cubes problem).
> >
> > We evaluate the agent based on whether its generated proof passes compilation. Passing confirms that the agent has made a reasonable, relevant, and formally correct attempt within strict logical constraints, even if it does not fully solve the open conjecture. This design is intentionally forward-looking, anticipating a future where autonomous agents can be directed to proactively explore such open-ended scientific frontiers.
> >
> > > Q3: In table 2, in the execution section, what does steps and time correspond to exactly? Are those obtained from human trajectories? Are those human expert or non-expert?
> >
> > Thanks for the question. One step is defined as one "plan/thought" generation plus its corresponding action (one ReACT) or a corresponding human action. The value (avg. steps) shown is the average number of steps used by different human annotators (literate but not experts in the specific tasks or software, detailed in your next question) to complete these tasks.
> >
> > 'Time consumption' is the average time taken by three mainstream commercial models to go through one evaluation run.
> >
> >
> > > Minor Part:
> > > How was the human baseline obtained? Was it through domain experts using the same interface as the agents?
> >
> > The human baseline was obtained by university students with STEM majors. To ensure a fair comparison with agents, while participants possessed general software/programming skills, they had no prior experience with the target science software and tools covered in ScienceBoard. As described in Appendix D.1 TASK ANNOTATION (Line 1055), they performed the tasks following official tutorials (Table 6) using the exact same interface as the agents.
> >
> > > Minor Part:
> > > line 376: Gemini-2.0-Flash (Team, 2024) -> (Google, 2024)
> >
> > For the Gemini-2.0-Flash (Team, 2024) -> (Google, 2024), thank you for pointing this out and it will be fixed during revision.
> >
> > Once again, we appreciate your recognition of our work and for your questions, which helped us significantly improve the paper. We hope that our response addresses your concerns.

---

> > > ### Comment · Reviewer_N2NY · 2025-11-22
> > > **Rebuttal response**
> > >
> > > Thank you for addressing the concerns I had raised. I'm glad to see you already thought about how to scale the benchmark up. I also saw in your other replies that you ran more up-to-date models which is great. I firmly believe this work is a timely benchmark to help push autonomous agents for scientific workflows. I've raised my score from 6 to 8.

---

> > > > ### Author Response · Authors · 2025-11-23
> > > > **Thanks**
> > > >
> > > > Dear Reviewer,
> > > >
> > > > We truly appreciate your recognition of our work and the value of our infrastructure. We will continue to refine the manuscript following your suggestions.
> > > >
> > > > Thank you once again for your insightful review and support!

---

### Official Review · Reviewer_cnsQ · 2025-10-31

**Soundness:** 3
**Presentation:** 1
**Contribution:** 2
**Rating:** 4
**Confidence:** 4

**Summary:**

The paper presents "yet another benchmark" for evaluating current LLM agents' capabilities in navigating typical scientific workflows across different domains. These workflows encompasses interacting with multiple tools, parsing feedback, multi-modal interpretation and reasoning, coding and domain-specific knowledge. The environment provided to an agent is VM with the necessary tools and the action space is unified through GUI and CLI. The conclusion is that current LLMs in the standard ReAct framework severely under-performs human across all domains (with overall success rates of ~15% vs ~60%). One interesting empirical finding is that separating planner and action executor improves the performance.

**Strengths:**

1. The paper presents a well-engineered agentic evaluation and a benchmark on realistic and complex scientific workflows across multiple domains.
2. The experimental analysis is quite comprehensive, with ablations on the operators of ReAct and tools.

**Weaknesses:**

1. There are no experiments with test-time scaling infra -- these tasks are too complex to be done in one-shot with simple prompting. Current experiments do not fully and fairly demonstrate the LLM abilities to reason and interact with environment since the provided compute infra is quite limited.
2. The sampling params like max_tokens = 1500 seem too restrictive for the frontier LLMs, also it seems that they were not run in reasoning mode. Even then, these LLMs typically would require a lot of think tokens (much more than 1.5k tokens) on these complex tasks.
3. Several technical details are unclear. For example, how is the memory implemented -- is it simply the concatenation of observations, actions or did it involve some summarization? With more complex tasks, the prompt can get super long and naturally LLMs will run into all sort of long-context issues. Also. please provide ReAct traces for a few tasks.
3. The presentation can be improved with better formatting on tables, so that one can easily get some insights on what models perform well on certain domains.

**Questions:**

Check above weaknesses.
1. I wonder if the models were run in native-reasoning mode (gpt-oss even supports different level of reasoning).
2. Did the authors run ablations or sweeps on sampling parameters, especially max_tokens?

---

> ### Author Response · Authors · 2025-11-19
> **Response to Reviewer cnsQ (1/2)**
>
> > W1:  Current experiments do not fully and fairly demonstrate the LLM abilities to reason and interact with environment since the provided compute infra is quite limited.
>
> Thank you for raising this point. To address the concern about limited compute, we conducted an additional experiment scaling inference-time compute.
> We (1) extended max_tokens to 2500 and (2) evaluated GPT-5 with varying levels of test time reasoning effort across three distinct domains. The results are shown below:
>
> | | Algebra | Biochem | GIS |
> | :--- | :--- | :--- | :--- |
> | **Observation = screenshot** | | | |
> | *compute* | | | |
> | reasoning\_effort = medium | 41.90% | 65.52% | 14.70% |
> | reasoning\_effort = high | 45.16% | 65.52% | 14.70% |
> | **Observation = screenshot + a11ytree** | | | |
> | *compute* | | | |
> | reasoning\_effort = medium | 48.39% | 68.96% | 14.70% |
> | reasoning\_effort = high | 51.61% | 72.41% | 17.64% |
>
> As observed, performance improves to varying degrees with increased compute, confirming that the reasoning capability leads to a small performance enhancement.
>
> This result indicates that while scaling compute provides help, it does not resolve the fundamental difficulties. This aligns with our experimental insights in Section 5.2. The current bottlenecks likely stem more from *the agent's ability to comprehend the scientific task* (perceiving the science software environment and deriving an appropriate plan) and *make the correct actions* (e.g., precisely clicking the target element), rather than simply scaling compute.
>
>
> > W2 & Q2: The sampling params like max_tokens = 1500 seem too restrictive for the frontier LLMs. Did the authors run ablations or sweeps on sampling parameters, especially max_tokens?
>
> Thank you for this question. The original setting was primarily chosen to manage the high evaluation costs associated with commercial models.
>
> Following your request, we conduct a controlled ablation study on max_tokens using the Screenshots setting on three domains. The results are as follows:
>
> | max\_token | Algebra | Biochem | GIS |
> | :--- | :--- | :--- | :--- |
> | 1500 | 41.90% | 62.10% | 11.8% |
> | 2000 | 41.90% | 58.62% | 11.8% |
> | 2500 | 41.90% | 65.52% | 14.70% |
>
> We observe that increasing max_tokens has a negligible impact on Algebra tasks, but yields some performance improvement in the Biochemistry and GIS domains.
> We appreciate your valuable suggestions and hope these experiments address your concerns. While running a full sweep across domains is extremely resource-intensive due to the cost, we will expand this experiment and incorporate more comprehensive results in the final version.
>
> > W3: How is the memory implemented? With more complex tasks, the prompt can get super long and naturally LLMs will run into all sort of long-context issues.
>
> Memory Implementation: Our memory follows standard practices[1,2], where the history of reasoning, actions, and multimodal observations is concatenated.
>
> Handling Long Context: To address your valid concern about long-context issues (and cost), we employ a sliding window strategy. Both commercial and open-source models retain the context of the last 3 react history. This design is empirically grounded in findings from OSWorld [1] (specifically their analysis on "The effect of length of history on performance"), trying to strike a balance between context and performance.
>
> Furthermore, to mitigate your concerns about "super long prompts," we treat the accessibility tree (a11ytree) as optional during inference in a11ytree + screenshot setting. Since a11ytrees are often very redundant, removing them from history allows the model to operate efficiently within the context window.
>
> > W4: The presentation can be improved with better formatting on tables
>
> Thank you for your suggestion. Due to page limits in the  current submission, we have to compress the info in the table, moving many detailed insights to Appendix F. We will utilize the additional page (if available) to improve the table presentations and readability as advised.

---

> ### Author Response · Authors · 2025-11-19
> **Response to Reviewer cnsQ (2/2)**
>
> > Q1: I wonder if the models were run in native-reasoning mode
>
> Yes. For QvQ-72B-Preview and GPT-oss-120B covered in the experiments, we utilized their native reasoning modes.
> We observed that while QvQ-72B-Preview has strong reasoning capabilities, it achieved lower scores primarily because it struggled to strictly follow the required action output format, rather than a lack of reasoning ability.
>
> In addition to the compute-scaling experiments mentioned in W1, we conducted further evaluations using native-reasoning modes during this discussion period. The results are presented below:
>
> | Model | Algebra | Biochem | GIS | ATP | Astron | Doc |
> | :--- | :--- | :--- | :--- | :--- | :--- | :--- |
> | **A11ytree + Screenshots** | | | | | | |
> | GPT-5 | 41.9% | 62.1% | 11.8% | 7.7% | 15.2% | 6.25% |
> | Claude-4-Sonnet | 48.4% | 65.5% | 17.6% | 3.84% | 21.2% | 25.0% |
> | Gemini-2.5-Pro | 38.7% | 55.2% | 17.6% | 0.0% | 18.2% | 12.5% |
> | **Screenshots** | | | | | | |
> | GPT-5 | 12.90% | 31.0% | 5.88% | 0.00% | 12.1% | 12.5% |
> | Claude-4-Sonnet | 22.6% | 17.2% | 14.7% | 3.84% | 18.20% | 6.25% |
> | Gemini-2.5-Pro | 9.7% | 31.0% | 2.94% | 0.00% | 9.09% | 12.5% |
>
> > Q2: is answered in the previous block.
>
> > W3 Cont'd: Also. please provide ReAct traces for a few tasks.
>
> Thank you for this request. In response, we present a ReAct trace for a biochem task below.
>
> Task: Add hydrogens with method of 'steric only' in ChimeraX.
>
> ## Step1:
>
> [Screenshot 1]
>
> I'm looking at a ChimeraX session with a molecular model loaded (labeled as "4r0u"). I can see the ChimeraX interface with various menu items including File, Edit, Select, Actions, Tools, etc. There's also a command line interface at the bottom labeled "Command:" which will be useful for our task.
>
> The task is to add hydrogens with the method of 'steric only' in ChimeraX. In ChimeraX, this can be done through the command line interface by using the "addh" command with the appropriate option.
>
> I'll click on the command line interface and enter the appropriate command to add hydrogens with the steric only method.
>
> ```python
> # Click on the command line interface
>
> pyautogui.click(750, 644)  \# Click on the command line area
>
> # Type the command to add hydrogens with steric only method
>
> pyautogui.write("addh steric true")
>
> # Press Enter to execute the command
> pyautogui.press('enter')
> ```
>
> ## Step2:
>
> [Screenshot 2]
>
> I can see that my previous command attempt resulted in an error - "Expected a keyword" is shown in the log, indicating the syntax I used wasn't correct. Looking at the command line, I tried "addh steric true" but that's not the correct syntax for the command.
>
> In ChimeraX, the correct syntax for adding hydrogens with the steric only method is to use the "method" keyword. Let me try again with the proper command syntax.
>
> ```python
> # Click on the command line interface
>
> pyautogui.click(500, 966)  # Click on the command line area
>
> # Clear the current text
>
> pyautogui.hotkey('ctrl', 'a')  # Select all
>
> pyautogui.press('delete')  # Delete the text
>
> # Type the correct command to add hydrogens with steric only method
>
> pyautogui.write("addh method steric")
>
> # Press Enter to execute the command
> pyautogui.press('enter')
> ```
> ...
>
> ## Step5:
>
> [Screenshot 5]
>
> Looking at the current state of the ChimeraX session and the log information, I can see that the task has been successfully completed. The command "addh hbond false" has been executed, and the log shows "Summary of feedback from adding hydrogens to 4r0u #1" with details including "61 hydrogens added".
>
> The molecular structure now shows the added hydrogens visibly in the 3D view, appearing as small white spheres attached to the molecule. The system has processed the command successfully and completed the hydrogen addition using the appropriate methodology.
>
> Since the task of adding hydrogens with the method of 'steric only' (which was accomplished by using "hbond false" parameter) has been completed successfully, I'll return the special code indicating task completion:
>
> ```python
> DONE
> ```
>
> Due to the lack of image support on OpenReview, the tag [Screenshot] within the trace represents the visual information available to the agent at that step.
>
> We thank you again for your review. We hope these additional experiments address your concerns, and we look forward to hearing from you.

---

> > ### Comment · Reviewer_cnsQ · 2025-11-19
> > **Rebuttal response**
> >
> > Dear authors,
> >
> > Thank you for addressing my concerns.
> > I do acknowledge the challenges in setting up infra for such large-scale expts which are extremely resource-consuming and costly. I appreciate the additional expts and case studies -- these provide valuable insights. Please add them in the revised paper.
> >
> > I've raised my score from 4 to 6. Good luck!

---

> > > ### Author Response · Authors · 2025-11-20
> > > **Thanks**
> > >
> > > Dear reviewer,
> > >
> > > We are delighted to hear that our response addressed your concerns! We will incorporate the additional experiments conducted during the discussion period and case studies into the revised paper.
> > >
> > > Thank you once again for your insightful review and support.

---

> ### Author Response · Authors · 2025-11-28
> **Thanks again for your feedback**
>
> Dear Reviewer,
>
> We have completed the additional large-scale experiments and will update the manuscript to include them.
>
> Thank you once again for your feedback and updated assessment.
>
> Best,
> Authors

---

### Official Review · Reviewer_CtCn · 2025-11-01

**Soundness:** 3
**Presentation:** 3
**Contribution:** 2
**Rating:** 6
**Confidence:** 3

**Summary:**

This paper presents ScienceBoard, a new benchmark for testing AI agents on real scientific software. It runs in a virtual machine and includes 169 tasks across six domains (like biochemistry and astronomy) using professional tools like ChimeraX and Celestia. Agents have to use both GUI and CLI to complete tasks. The authors tested top models like GPT-4o and found they fail badly, with only a 15% success rate, showing these agents aren't ready to be "AI co-scientists" yet.

**Strengths:**

The main strength is that this is a novel and needed benchmark. Testing agents on real scientific software is a great idea. The technical work to get this running is impressive, especially the way they check the internal state of the apps to see if a task is done. The paper is clearly written. The analysis is also good, especially the part where they separate the "planner" (like GPT-4o) from the "actor" (GUI models), which gives a good hint at how to build better agents.

**Weaknesses:**

The paper is strong, but a few things could be better. The pass/fail scoring is a bit harsh; an agent that almost finishes a task gets the same zero as one that fails immediately, so it's hard to see partial progress. The number of tasks per domain is a bit uneven. Also, the "scientific discovery" framing is a slight overstatement. The tasks are more about using scientific tools correctly, not really about discovering new science, which is fine but good to be clear about. The two "Open Problems" sound cool, but it's a shame there are only two.

**Questions:**

Your GUI-only vs. GUI+CLI analysis is great. Can you give a clearer breakdown of the 169 tasks? How many must be solved with the GUI (like in Celestia?), and how many are hybrid, where you disabled the CLI for the test? This would help clarify the balance of skills needed.

What are the two "Open Problems" mentioned in Table 2? They sound really interesting. How are they different from the "Hard" tasks, and how do you evaluate them?

I know the pass/fail scoring is tough. Since your tasks seem to have multiple steps (from Fig 3), have you thought about a partial credit or step-based success rate? It could show more granular progress, even if it's not the main metric.

---

> ### Author Response · Authors · 2025-11-19
> **Response to Reviewer CtCn (1/2)**
>
> Thank you for your encouraging review! We appreciate your recognizing our work as a great idea, the technical contribution of ScienceBoard, and the novelty of our benchmark. We decompose your comments and attempt to address them as follows:
>
> > W1 & Q3: The paper is strong, but a few things could be better. The pass/fail scoring is a bit harsh; an agent that almost finishes a task gets the same zero as one that fails immediately, so it's hard to see partial progress.
>
> Thank you for this insightful observation. We discussed this in our Limitations section (Line 849). We agree pass/fail is harsh, but defining reliable partial progress for open-ended, OS-level tasks across diverse software is extremely difficult due to the vast state spaces.
>
> Using VLM-Judges, the main alternative, introduces significant judge bias and known insensitivity to granular UI-level actions. To further address your concern, here we conduct additional experiments: In our own trial sampling 100 trajectories (by Claude, Gemini, QwenVL agent) with a GPT-4o judge, we found its accuracy was not yet mature enough:
>
> | | State-based Evaluation (ours) | VLM-Judge |
> | :--- | :--- | :--- |
> | ChimeraX | 100% | ~68% |
> | Kalgebra | 100% | ~74% |
>
> Therefore, while harsh, we chose the state-based (pass/fail) evaluation because it is robust, objective, and reproducible, aligning with other well-established benchmarks' approaches [1,2,3].
>
> > W2: The number of tasks per domain is a bit uneven.
>
> Thank you for this observation. The distribution across primary scientific domains is relatively balanced (Figure 4): Algebra (18.5%), Biochemistry (17.3%), Astronomy (19.6%), GIS (20.2%), and ATP (15.5%).
> The scientific documentation category (8.9%) is intentionally smaller, as it's a subset of tasks drawn from the other domains (i.e., report writing post-exploration). Minor deviations in the main domains exist because we analyzed each software's exploration space and pruned similar tasks to prioritize diversity over raw numbers.
>
> > W3: The tasks are more about using scientific tools correctly, not really about discovering new science, which is fine but good to be clear about.
>
> Thank you for the observation. We place much effort into realizing automation of the scientific workflow. We'd like to note this workflow requires not just tool / software use, but also domain knowledge (formal math, biology, geography), programming, complex scientific reasoning, data processing, and writing.
>
> We agree the "new discovery" aspect is limited, as this is extremely difficult to evaluate programmatically. However, we do include two open-ended problems specifically to touch upon this, which we will detail in our answer to your next question.
>
> > W4 & Q2: The two "Open Problems" sound cool, but it's a shame there are only two. What are the two "Open Problems" mentioned in Table 2? They sound really interesting. How are they different from the "Hard" tasks, and how do you evaluate them?
>
> These two "Open Problems" are our specific attempt to address "discovering new science" (as mentioned in your W3). The key difference is that 'Hard' tasks have a defined ground truth, whereas 'Open Problems' do not. A typical example is asking the agent to use all available tools and knowledge to prove the "Sum of Four Cubes problem."
> We use formal methods: `def OP_1 (n : ℤ) : Decidable (∃ a b c d : ℤ, n = a ^ 3 + b ^ 3 + c ^ 3 + d ^ 3) := by sorry`. to convert the challenge into a deterministic, programmatic one.
> We can then judge the correctness of the agent's problem-solving approach by whether its generated proof passes compilation. Even if the agent does not fully solve the open conjecture, passing this confirms that it has at least made a reasonable, relevant, and formally correct attempt within strict logic.
>
> The design of these problems is intentionally forward-looking. We envision a future where autonomous agents can be directed to proactively explore such open-ended scientific questions.
>
> ### References
>
> [1] OSWorld: Benchmarking Multimodal Agents for Open-Ended Tasks in Real Computer Environments https://arxiv.org/abs/2404.07972
>
> [2] AndroidWorld: A Dynamic Benchmarking Environment for Autonomous Agents https://arxiv.org/abs/2405.14573
>
> [3] WebArena: A Realistic Web Environment for Building Autonomous Agents https://arxiv.org/abs/2307.13854

---

> > ### Author Response · Authors · 2025-11-19
> > **Response to Reviewer CtCn (2/2)**
> >
> > > Q1: Your GUI-only vs. GUI+CLI analysis is great. Can you give a clearer breakdown of the 169 tasks? How many must be solved with the GUI (like in Celestia?), and how many are hybrid, where you disabled the CLI for the test? This would help clarify the balance of skills needed.
> >
> >
> > Thank you for your interest in our analysis. Regarding the breakdown:
> >
> > - GUI-only: Celestia (~19.6%) is strictly GUI-only due to software limitations on CLI.
> >
> > - Hybrid: The remaining ~80.4% (ChimeraX, KAlgebra, GRASS GIS, TeXStudio, Lean) support both interfaces.
> >
> > In hybrid settings, agents show distinct preferences (e.g., Claude favors CLI, InternVL favors GUI).
> > For the specific ablation study, we chose ChimeraX to ensure fairness. Unlike other software where GUI and CLI actions may not map equivalently, ChimeraX allows nearly all functions (including advanced settings) to be accessed via commands. This functional parity ensured that the observed performance differences were due to the interaction mode, not feature availability.
> >
> > Once again, we thank you for your review and the recognition of our work. We sincerely hope that our explanations and additional experiments can address your concerns.

---

### Official Review · Reviewer_fF65 · 2025-11-01

**Soundness:** 2
**Presentation:** 2
**Contribution:** 2
**Rating:** 4
**Confidence:** 4

**Summary:**

This work presents ScienceBoard, a multi-modal benchmark for evaluating computer-use agents in automating scientific workflows. The benchmark covers 169 expert-curated tasks from several scientific domains and focuses on the agents' abilities in using domain-specific software to complete scientific tasks. Evaluations results on several popular agents reveal the challenge of existing models to solve the tasks.

**Strengths:**

1. This work is a timely addition to the community for evaluation and development of autonomous agents for scientific workflows.
2. Evaluation results delivers insights into the performance of existing agents, especially under different input modalities.

**Weaknesses:**

1. I think the tested models are outdated for this conference. Given the rapid development of the field, it will be great to include results from more recent models, such as GPT-5, Gemini-2.5, Qwen3, etc.
2. Related to 1, the cost of evaluation is unclear from the paper, which may hinder the adoption.
3. Several key details are missing, which are critical for assessing the quality of the benchmark:
    1. What are the background of the experts/annotators? Since the tasks are drafted by annotators, are they ecologically valid tasks that correspond to real-world scientific workflows?
    2. line 325, how is difficulty defined?
    3. How is human performance measured? Intuitively, average population and professionals should perform quite differently for scientific tasks, it is important to discuss the background of participants in human study.
4. The scope of this paper seems to be over-claimed.  While the author claims to cover "end-to-end scientific exploration workflows", critical research steps like literature review, idea generation, and report/paper writing are missing. The focus is mainly on execution and using scientific software.
5. Evaluation setup is unclear. How many trials are allowed for each agent? Do you use the same context length limit for all agents? Also, as far as I know, o3-mini supports multimodal input. Why it is only evaluated in the a11ytree only setting?

**Questions:**

1. Table 2, what's the unit for length? How is time consumption calculated?

---

> ### Author Response · Authors · 2025-11-19
> **Response to Reviewer fF65 (1/2)**
>
> Thank you for your review! We appreciate your recognizing our work as a timely addition to the community and that our evaluation results deliver insights. We summarize and address your concerns as follows:
>
> > W1:  Given the rapid development of the field, it will be great to include results from more recent models, such as GPT-5, Gemini-2.5, Qwen3, etc.
>
> As requested, we have run additional experiments to evaluate the performance of more recent models on ScienceBoard (under both Screenshot and Screenshot + a11ytree settings).
>
> | Model | Algebra | Biochem | GIS | ATP | Astron | Doc |
> | :--- | :--- | :--- | :--- | :--- | :--- | :--- |
> | **A11ytree + Screenshots** | | | | | | |
> | GPT-5 | 41.9% | 62.1% | 11.8% | 7.7% | 15.2% | 6.25% |
> | Claude-4-Sonnet | 48.4% | 65.5% | 17.6% | 3.84% | 21.2% | 25.0% |
> | Gemini-2.5-Pro | 38.7% | 55.2% | 17.6% | 0.0% | 18.2% | 12.5% |
> | **Screenshots** | | | | | | |
> | GPT-5 | 12.90% | 31.0% | 5.88% | 0.00% | 12.1% | 12.5% |
> | Claude-4-Sonnet | 22.6% | 17.2% | 14.7% | 3.84% | 18.20% | 6.25% |
> | Gemini-2.5-Pro | 9.7% | 31.0% | 2.94% | 0.00% | 9.09% | 12.5% |
>
> The experimental results confirm that these newer models indeed achieve higher scores on the benchmark. We will incorporate these findings into the revision to ensure our paper is timely and up-to-date.
>
> > W2: Related to 1, the cost of evaluation is unclear from the paper, which may hinder the adoption.
>
> Thank you for the question. We have already provided the evaluation costs for commercial models in Appendix D.7 EVALUATION COST, located at Line 1222 of the current submission version. Typically, we report the average cost among different observations.
>
> | Model | GPT-4o | Claude-3.7-Sonnet | Gemini-2.0-Flash | Gemini-2.5-Pro |
> | :--- | :--- | :--- | :--- | :--- |
> | Pricing | $64 | $86 | $45 | $177 |
>
> These API prices are subject to change by the providers. We will also update the final version to include the costs of the new experiments conducted during this Author-Reviewer discussion period.
>
> > W3: (1) background of the experts/annotators? (2) How is human performance measured? (3) How is difficulty defined?
>
> Annotator Background: The annotation was performed by university students with STEM majors. While they possessed software / programming skills, they had no prior experience with the specific software / tools covered in ScienceBoard. As described in Appendix D.1 TASK ANNOTATION (Line 1055), the annotation process followed official software tutorials (Table 6), corresponding to real-world scientific workflows. We also verified the task distribution in Appendix D.2 TASK DIVERSITY (Line 1063).
>
> Human Performance: This is detailed in Appendix D.5 HUMAN PERFORMANCE (Line 1189). We recruited college-level students to establish this baseline. Before attempting the tasks, participants were required to familiarize themselves with the relevant scientific knowledge and study the provided operational manuals/tutorials. We then recorded the time taken, the average number of steps, and their success (correctness) for each task.
>
> Difficulty Definition: Difficulty is defined by the avg. number of human steps measured in the previous step. The levels correspond to 1-5 steps (Easy), 5-10 steps (Medium), and >10 steps (Hard). Our analysis in Appendix F.3 DIFFICULTY ANALYSIS (Line 1389) confirms that this division effectively measures the tiered capabilities of different models.
>
> > W4: While the author claims to cover "end-to-end scientific exploration workflows", critical research steps like literature review, idea generation, and report/paper writing are missing.
>
> Thank you for this point! Firstly, regarding report/paper writing, we designed writing tasks (Sec 4.1 (d)) specifically to evaluate an agent's ability to produce reports after completing experiments / exploration.
>
> Then, we agree that literature review and idea generation are essential steps toward a full "AI scientist." While, this work's core focus and novelty is on multimodal autonomous agents that bridge the gap from conceptualization to Experiment / Implementation [1], a phase we consider a cornerstone for full automation.
>
> We currently do not cover those steps because: (1) They are highly open-ended, making reliable evaluation extremely difficult, unlike the deterministic tasks in ScienceBoard. (2) Integrating them with our complex, multimodal workflow would very likely exceed current model capacity. Notwithstanding, these steps can be modular and our environment can serve as the engine to bridge a larger pipeline.
>
> While we cannot cover such a big scope in a single work, we sincerely appreciate this insightful comment. You have accurately outlined the vision for a complete research pipeline, which provides a valuable direction for future work as models progress toward AGI.
>
> ### Reference
>
> [1] Llm4sr: A survey on large language models for scientific research, 2025

---

> ### Author Response · Authors · 2025-11-19
> **Response to Reviewer fF65 (2/2)**
>
> > W5: Evaluation setup is unclear. How many trials are allowed for each agent? Do you use the same context length limit for all agents? Also, as far as I know, o3-mini supports multimodal input. Why it is only evaluated in the a11ytree only setting?
>
> Thanks for the question. Each agent is evaluated in a single trial (we provide a stability analysis in Appendix D.6). Yes, the length limit is consistent across all agents. To ensure we stay within limits while maintaining history (up to 3 turns), we adapted the a11ytree setting to filter out non-essential elements.
>
> Regarding o3, as you requested, we have now evaluated its performance with multimodal input (screenshots + a11ytree). The results are as follows:
>
> | Model | Algebra | Biochem | GIS | ATP | Astron | Doc |
> | :--- | :--- | :--- | :--- | :--- | :--- | :--- |
> | o3 | 22.6% | 41.4% | 14.7% | 3.84% | 9.1% | 12.5% |
>
> > Q1: Table 2, what's the unit for length? How is time consumption calculated?
>
> For 'Length' in Table 2, the unit is a single step of agent / human interaction with the system, defined as one "plan/thought" generation plus its corresponding action for agents (one ReACT). The value shown is the average number of steps required by different human annotators to complete these tasks. 'Time consumption' is the average time taken by three mainstream commercial models to go through one evaluation run. All models were tested on identically configured Ubuntu VMs (please refer to our attached code for hardware details). We will add more information about this in the appendix.
>
> We thank you again for your questions, which help improve our paper. We hope our supplementary experiments and explanations can address your concerns.

---

> ### Author Response · Authors · 2025-11-25
> **Gentle Follow-up on Author - Reviewer Discussion**
>
> Dear Reviewer fF65,
>
> We have included additional experiments as per your request, and responded to the questions you raised. We would like to know if the explanations and results we provided can address your concerns.
>
> Thank you for your time and thoughtful questions.
>
> Best,
>
> The Authors

---

> > ### Comment · Reviewer_fF65 · 2025-11-25
> >
> > Thank you for the detailed response. I'm glad to see the up-to-date experiments and have raised the score from 4 to 6. I strongly encourage you to incorporate critical details into the main text and add pointers to corresponding appendices in the revision for better clarity. Good luck.

---

> > > ### Author Response · Authors · 2025-11-28
> > > **Thanks again for your feedback**
> > >
> > > Dear Reviewer,
> > >
> > > We are delighted to hear that our new experiments addressed your concerns and that you have updated your rating!
> > >
> > > We will certainly follow your suggestion to incorporate details into the main text and add pointers to the corresponding appendices in the revision.
> > >
> > > Thank you once again for your insightful review and continued support.

---

### Comment · Area_Chair_R7Jx · 2025-11-22

Dear Authors and Reviewers,

I would like to thank the authors for providing detailed rebuttal messages on time.

To reviewers: I would like to encourage you to carefully read all other reviews and the author responses and engage in an open exchange with the authors. Please post your first response as soon as possible within the discussion time window. Ideally, all reviewers will respond to the authors, so that the authors know their rebuttal has been read.

Best regards,
AC

---

### Author Response · Authors · 2025-12-03
**General Response and Summary of Rebuttal Updates**

Dear Area Chair(s),

We sincerely appreciate the additional efforts invested in handling the submissions under the current circumstances. Here, we summarize below the key facts regarding our submission and the constructive author–reviewer discussion, which we hope will assist you in making recommendations. We confirm that we have engaged in the discussion strictly in accordance with the ICLR Code of Conduct.

During the previous discussion phase, we actively participated in conversations with reviewers and received positive responses / comments from 3 of the 4 reviewers. As a result, the overall score improved significantly from `4, 6, 4, 6` to `6, 6, 6, 8` at the early stage of author-reviewer discussion, well before the incident. We summarize the discussion highlights in the table below.

| Reviewer | Initial Rating | Concerns / Request                                                                                                                  | Responses / Extra Exps                                                                                   | Updated Rating |
|----------|----------------|-------------------------------------------------------------------------------------------------------------------------------------|----------------------------------------------------------------------------------------------------------|----------------|
| `fF65`     | 4              | Tested models are old; evaluation cost is unclear; missing some annotator details.                                                  | Added experiments with GPT-5, Gemini-2.5-pro, Claude-4; clarified pricing & annotator background.        | **6 (Score Updated)**            |
| `CtCn`     | 6              | Pass/fail metric seems harsh; clarify "Open Problems" and task distribution.                                                        | Justified state-based eval vs. VLM-judges (with stats); explained formal verification for Open Problems; provide task distribution details. | 6 (No Reply)   |
| `cnsQ`     | 4              | Compute/token limits too restrictive; request ReAct traces and native reasoning mode experiments.                                   | Scaled compute & max_tokens; additional evaluations on reasoning models; provided full ReAct traces.     | **6 (Score Updated)**             |
| `N2NY`     | 6              | Want Clarification on Train/Test/Eval split; request deeper GUI vs. CLI performance analysis; question about infra/benchmark safety | Clarified evaluation focus; added more analysis on GUI/CLI dynamics and virtual machine safeguards.      | **8 (Score Updated)**             |

Discussion Outcomes and Progress:

1. **Positive Consensus & Updated Ratings**: Reviewers made positive responses and raised their scores significantly ( `fF65` `4->6`, `cnsQ` `4->6`, and `N2NY` `6->8`) after reviewing our rebuttal. They acknowledged that our (1) new experiments on SOTA models, (2) new experiments on reasoning models with more compute and (3) detailed clarifications, effectively addressed their concerns regarding model recency, compute scaling, and infra / evaluation.

2. **Address Remaining Concerns**: Although Reviewer `CtCn` did not respond to the final thread, given the initial positive score (6), we believe our response can resolve concerns regarding the "harshness" of the pass/fail metric and further strengthen the logic of building rule-based evaluators. We additionally provided comparative experiments showing that our state-based evaluation is more objective and robust than VLM-judge alternatives for scientific + computer-using tasks.

Once again, we thank the ACs for your time and patience. We hope this summary provides a clear picture of the essence of the entire constructive discussion phase. We kindly hope that you consider these positive discussion outcomes, specifically the consensus and the significant rating improvements (`avg. 5 -> 6.5`) before the security incident & rollback, when making your recommendation! For the detailed explanation and full experiments conducted, please kindly review our discussion with each reviewer.

At last, we remain fully committed to incorporating all the valuable insights gathered during this discussion period into our final revision to further elevate the quality and value of our work.

Best regards,

Authors of Submission 565

---

### Meta-Review · Area_Chair_UZmo · 2025-12-29

**Summary:**

The paper introduces a timely, well-engineered benchmark for evaluating multimodal, computer-using agents in realistic scientific workflows. The core contribution is building and validating a VM-based benchmark with real scientific software, domain diversity, and programmatic evaluation.

**Reviewer Concerns:**

The initial skepticism from the reviewers was substantially alleviated through additional experiments, clarifications, and rebuttals. Some common concerns are listed as below.
(i) Overly broad or overstated scope claims (e.g., "scientific discovery", "end-to-end workflows");
(ii) Insufficient clarity on evaluation methodology and setup;
(iii) Fairness and completeness of model evaluation (compute, tokens, reasoning modes);
(iv) Evaluation metrics being too coarse (pass or fail only);
(v) Need for stronger transparency and reproducibility details;
(vi) Some presentation and readability issues.

The author has actively participated in conversations with reviewers and received positive responses. The reviewers have reached the consensus of accepting this paper. All major objections were addressed.  I suggest the author to consider the below improvements: (i) Move critical methodological details into the main text with explicit appendix pointers. (ii) Improve table readability and clarity. (iii) Tone down "scientific discovery" claims and sharpen positioning around scientific tool-use and execution.(iv) Explicitly document evaluation setup, costs, human baselines, and difficulty definitions.

**Reviewer Scores:**

fF65: 6

CtCn: 6

cnsQ: 6

N2NY: 8

---

### Decision · Program_Chairs · 2026-01-26

Accept (Poster)